# Learning to Retrieve Reasoning Paths over Wikipedia Graph for Question Answering

**Akari Asai**[*†]**, Kazuma Hashimoto**[‡]**, Hannaneh Hajishirzi**[†§]**, Richard Socher**[‡] **& Caiming Xiong**[‡]

[†]University of Washington    [‡]Salesforce Research    [§]Allen Institute for Artificial Intelligence
{akari,hannaneh}@cs.washington.edu
{k.hashimoto,rsocher,cxiong}@salesforce.com

## Abstract

Answering questions that require multi-hop reasoning at web-scale necessitates retrieving multiple evidence documents, one of which often has little lexical or semantic relationship to the question. This paper introduces a new graph-based recurrent retrieval approach that learns to retrieve reasoning paths over the Wikipedia graph to answer multi-hop open-domain questions. Our retriever model trains a recurrent neural network that learns to sequentially retrieve evidence paragraphs in the reasoning path by conditioning on the previously retrieved documents. Our reader model ranks the reasoning paths and extracts the answer span included in the best reasoning path. Experimental results show state-of-the-art results in three open-domain QA datasets, showcasing the effectiveness and robustness of our method. Notably, our method achieves significant improvement in HotpotQA, outperforming the previous best model by more than 14 points.[1]

## 1 Introduction

Open-domain Question Answering (QA) is the task of answering a question given a large collection of text documents (e.g., Wikipedia). Most state-of-the-art approaches for open-domain QA (Chen et al., 2017; Wang et al., 2018a; Lee et al., 2018; Yang et al., 2019) leverage non-parameterized models (e.g., TF-IDF or BM25) to retrieve a fixed set of documents, where an answer span is extracted by a neural reading comprehension model. Despite the success of these pipeline methods in single-hop QA, whose questions can be answered based on a single paragraph, they often fail to retrieve the required evidence for answering multi-hop questions, e.g., the question in Figure 1. Multi-hop QA (Yang et al., 2018) usually requires finding more than one evidence document, one of which often consists of little lexical overlap or semantic relationship to the original question. However, retrieving a fixed list of documents independently does not capture relationships between evidence documents through *bridge entities* that are required for multi-hop reasoning.

Recent open-domain QA methods learn end-to-end models to jointly retrieve and read documents (Seo et al., 2019; Lee et al., 2019). These methods, however, face challenges for entity-centric questions since compressing the necessary information into an embedding space does not capture lexical information in entities. Cognitive Graph (Ding et al., 2019) incorporates entity links between documents for multi-hop QA to extend the list of retrieved documents. This method, however, compiles a fixed list of documents independently and expects the reader to find the reasoning paths.

In this paper, we introduce a new recurrent graph-based retrieval method that learns to retrieve evidence documents as reasoning paths for answering complex questions. Our method sequentially retrieves each evidence document, given the history of previously retrieved documents to form several reasoning paths in a graph of entities. Our method then leverages an existing reading comprehension model to answer questions by ranking the retrieved reasoning paths. The strong interplay between the retriever model and reader model enables our entire method to answer complex questions by exploring more accurate reasoning paths compared to other methods.

---

[*]Work partially done while the author was a research intern at Salesforce Research.
[1]Our code and data id available at https://github.com/AkariAsai/learning_to_retrieve_reasoning_paths.

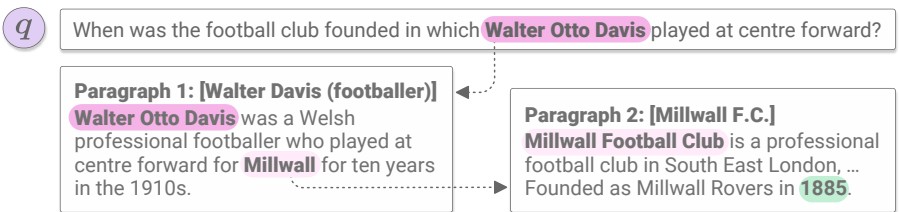

Figure 1: An example of open-domain multi-hop question from HotpotQA. Paragraph 2 is unlikely to be retrieved using TF-IDF retrievers due to little lexical overlap to the given question.

To be more specific, our method (sketched in Figure 2) constructs the Wikipedia paragraph graph using Wikipedia hyperlinks and document structures to model the relationships between paragraphs. Our retriever trains a recurrent neural network to score reasoning paths in this graph by maximizing the likelihood of selecting a correct evidence paragraph at each step and fine-tuning paragraph BERT encodings. Our reader model is a multi-task learner to score each reasoning path according to its likelihood of containing and extracting the correct answer phrase. We leverage data augmentation and negative example mining for robust training of both models.

Our experimental results show that our method achieves the state-of-the-art results on HotpotQA full wiki and HotpotQA distractor settings (Yang et al., 2018), outperforming the previous state-of-the-art methods by more than 14 points absolute gain on the full wiki setting. We also evaluate our approach on SQuAD Open (Chen et al., 2017) and Natural Questions Open (Lee et al., 2019) without changing any architectural designs, achieving better or comparable to the state of the art, which suggests that our method is robust across different datasets. Additionally, our framework provides interpretable insights into the underlying entity relationships used for multi-hop reasoning.

## 2 RELATED WORK

**Neural open-domain question answering**  Most current open-domain QA methods use a pipeline approach that includes a *retriever* and *reader*. Chen et al. (2017) incorporate a TF-IDF-based retriever with a state-of-the-art neural reading comprehension model. The subsequent work improves the heuristic retriever by re-ranking retrieved documents (Wang et al., 2018a;b; Lee et al., 2018; Lin et al., 2018). The performance of these methods is still bounded by the performance of the initial retrieval process. In multi-hop QA, non-parameterized retrievers face the challenge of retrieving all the relevant documents, one or some of which are lexically distant from the question. Recently, Lee et al. (2019) and Seo et al. (2019) introduce fully trainable models that retrieve a few candidates directly from large-scale Wikipedia collections. All these methods find evidence documents independently without the knowledge of previously selected documents or relationships between documents. This would result in failing to conduct multi-hop retrieval.

**Retrievers guided by entity links**  Most relevant to our work are recent studies that attempt to use entity links for multi-hop open-domain QA. Cognitive Graph (Ding et al., 2019) retrieves evidence documents offline, and trains a reading comprehension model to jointly predict possible answer spans and next-hop spans to extend the reasoning chain. Instead, we train our retriever to find reasoning paths directly. Concurrent with our work, Entity-centric IR (Godbole et al., 2019) uses entity linking for multi-hop retrieval. Unlike our method, this method does not learn to retrieve reasoning paths sequentially, nor study the interplay between retriever and reader. Moreover, while the previous approaches require a system to encode all possible nodes, our beam search decoding process only encodes the nodes on the reasoning paths, which significantly reduces the computational costs. PullNet (Sun et al., 2019) learns to retrieve question-aware sub-graphs from text corpora and knowledge bases (e.g., Freebase), while we focus on open-domain QA solely based on text.

**Multi-step (iterative) retrievers**  Similar to our recurrent retriever, multi-step retrievers explore multiple evidence documents iteratively. Multi-step reasoner (Das et al., 2019) repeats the retrieval process for a *fixed* number of steps, interacting with a reading comprehension model by reformulating the query in a latent space to enhance retrieval performance. Feldman & El-Yaniv (2019) also propose a query reformulation mechanism with a focus on multi-hop open-domain QA. Most recently, Qi et al. (2019) introduce GoldEn Retriever, which reads and generates search queries for two steps to search documents for HotpotQA full wiki. These methods do not use the graph

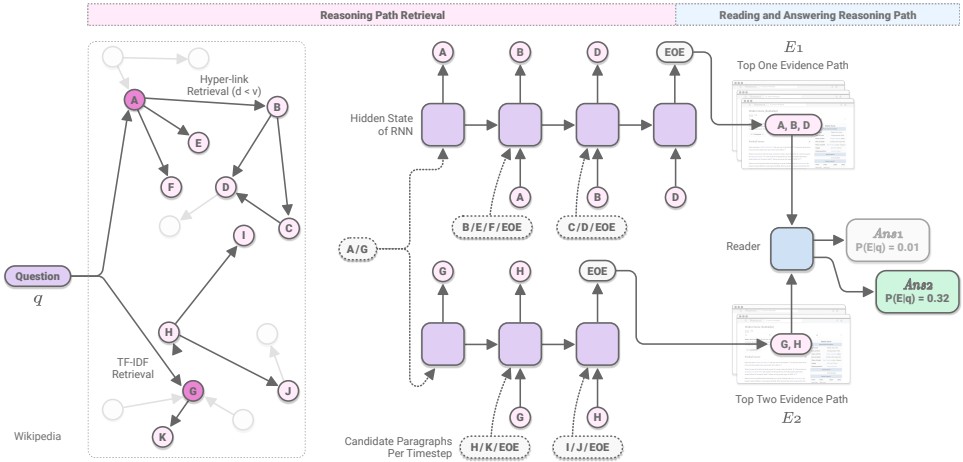

Figure 2: Overview of our framework.

structure of the documents during the iterative retrieval process. In addition, all of these multi-step retrieval methods do not accommodate arbitrary steps of reasoning and the termination condition is hard-coded. In contrast, our method leverages the Wikipedia graph to retrieve documents that are lexically or semantically distant to questions, and is adaptive to any reasoning path lengths, which leads to significant improvement over the previous work in HotpotQA and SQuAD Open.

## 3 OPEN-DOMAIN QUESTION ANSWERING OVER WIKIPEDIA GRAPH

**Overview** This paper introduces a new graph-based recurrent retrieval method (Section 3.1) that learns to find evidence documents as reasoning paths for answering complex questions. We then extend an existing reading comprehension model (Section 3.2) to answer questions given a collection of reasoning paths. Our method uses a strong interplay between retrieving and reading steps such that the retrieval method learns to retrieve a set of reasoning paths to narrow down the search space for our reader model, for robust pipeline process. Figure 2 sketches the overview of our QA model.

We use Wikipedia for open-domain QA, where each article is divided into paragraphs, resulting in millions of paragraphs in total. Each paragraph $p$ is considered as our retrieval target. Given a question $q$, our framework aims at deriving its answer $a$ by retrieving and reading reasoning paths, each of which is represented with a sequence of paragraphs: $E = [p_i, \ldots, p_k]$. We formulate the task by decomposing the objective into the retriever objective $S_{\mathrm{retr}}(q, E)$ that selects reasoning paths $E$ relevant to the question, and the reader objective $S_{\mathrm{read}}(q, E, a)$ that finds the answer $a$ in $E$:

$$\arg\max_{E,a} S(q, E, a) \quad \text{s.t.} \quad S(q, E, a) = S_{\mathrm{retr}}(q, E) + S_{\mathrm{read}}(q, E, a). \tag{1}$$

### 3.1 LEARNING TO RETRIEVE REASONING PATHS

Our method learns to retrieve reasoning paths across a graph structure. Evidence paragraphs for a complex question do not necessarily have lexical overlaps with the question, but one of them is likely to be retrieved, and its entity mentions and the question often entail another paragraph (e.g., Figure 1). To perform such multi-hop reasoning, we first construct a graph of paragraphs, covering all the Wikipedia paragraphs. Each node of the Wikipedia graph $\mathcal{G}$ represents a single paragraph $p_i$.

**Constructing the Wikipedia graph** Hyperlinks are commonly used to construct relationships between articles on the web, usually maintained by article writers, and are thus useful knowledge resources. Wikipedia consists of its internal hyperlinks to connect articles. We use the hyperlinks to construct the direct edges in $\mathcal{G}$. We also consider symmetric within-document links, allowing a paragraph to hop to other paragraphs in the same article. The Wikipedia graph $\mathcal{G}$ is densely connected and covers a wide range of topics that provide useful evidence for open-domain questions. This graph is constructed offline and is reused throughout training and inference for any question.

### 3.1.1  THE GRAPH-BASED RECURRENT RETRIEVER

**General formulation with a recurrent retriever**   We use a Recurrent Neural Network (RNN) to model the reasoning paths for the question $q$. At the $t$-th time step ($t \geq 1$) our model selects a paragraph $p_i$ among candidate paragraphs $\mathbf{C}_t$ given the current hidden state $h_t$ of the RNN. The initial hidden state $h_1$ is independent of any questions or paragraphs, and based on a parameterized vector. We use BERT's [CLS] token representation (Devlin et al., 2019) to independently encode each candidate paragraph $p_i$ *along with* $q$.[2] We then compute the probability $P(p_i|h_t)$ that $p_i$ is selected. The RNN selection procedure captures relationships between paragraphs in the reasoning path by conditioning on the selection history. The process is terminated when [EOE], the end-of-evidence symbol, is selected, to allow it to capture reasoning paths with arbitrary length given each question. More specifically, the process of selecting $p_i$ at the $t$-th step is formulated as follows:

$$w_i = \text{BERT}_{[\text{CLS}]}(q, p_i) \in \mathbb{R}^d, \tag{2}$$

$$P(p_i|h_t) = \sigma(w_i \cdot h_t + b), \tag{3}$$

$$h_{t+1} = \text{RNN}(h_t, w_i) \in \mathbb{R}^d, \tag{4}$$

where $b \in \mathbb{R}^1$ is a bias term. Motivated by Salimans & Kingma (2016), we normalize the RNN states to control the scale of logits in Equation (3) and allow the model to learn multiple reasoning paths. The details of Equation (4) are described in Appendix A.1. The next candidate set $\mathbf{C}_{t+1}$ is constructed to include paragraphs that are linked from the selected paragraph $p_i$ in the graph. To allow our model to flexibly retrieve multiple paragraphs within $\mathbf{C}_t$, we also add $K$-best paragraphs other than $p_i$ (from $\mathbf{C}_t$) to $\mathbf{C}_{t+1}$, based on the probabilities. We typically set $K = 1$ in this paper.

**Beam search for candidate paragraphs**   It is computationally expensive to compute Equation (2) over millions of the possible paragraphs. Moreover, a fully trainable retriever often performs poorly for entity-centric questions such as SQuAD, since it does not explicitly maintain lexical information (Lee et al., 2019). To navigate our retriever in the large-scale graph effectively, we initialize candidate paragraphs with a TF-IDF-based retrieval and guide the search over the Wikipedia graph. In particular, the initial candidate set $\mathbf{C}_1$ includes $F$ paragraphs with the highest TF-IDF scores with respect to the question. We expand $\mathbf{C}_t$ ($t \geq 2$) by appending the [EOE] symbol. We additionally use a beam search to explore paths in the directed graph. We define the score of a reasoning path $E = [p_i, \ldots, p_k]$ by multiplying the probabilities of selecting the paragraphs: $P(p_i|h_1) \ldots P(p_k|h_{|E|})$. The beam search outputs the top $B$ reasoning paths $\mathbf{E} = \{E_1, \ldots, E_B\}$ with the highest scores to pass to the reader model i.e., $S(q, E, a) = S_{\text{read}}(q, E, a)$ for $E \in \mathbf{E}$.

In terms of the computational cost, the number of the paragraphs processed by Equation (2) is bounded by $\mathcal{O}(|\mathbf{C}_1| + B \sum_{t \geq 2} \overline{|\mathbf{C}_t|})$, where $B$ is the beam size and $\overline{|\mathbf{C}_t|}$ is the average size of $\mathbf{C}_t$ over the $B$ hypotheses.

### 3.1.2  TRAINING OF THE GRAPH-BASED RECURRENT RETRIEVER

**Data augmentation**   We train our retriever in a supervised fashion using evidence paragraphs annotated for each question. For multi-hop QA, we have multiple paragraphs for each question, and single paragraph for single-hop QA. We first derive a ground-truth reasoning path $g = [p_1, \ldots, p_{|g|}]$ using the available annotated data in each dataset. $p_{|g|}$ is set to [EOE] for the termination condition. To relax and stabilize the training process, we augment the training data with additional reasoning paths – not necessarily the shortest paths – that can derive the answer. In particular, we add a new training path $g_r = [p_r, p_1, \ldots, p_{|g|}]$ by adding a paragraph $p_r \in \mathbf{C}_1$ that has a high TF-IDF score and is linked to the first paragraph $p_1$ in the ground-truth path $g$. Adding these new training paths helps at the test time when the first paragraph in the reasoning path does not necessarily appear among the paragraphs that initialize the Wikipedia search using the heuristic TF-IDF retrieval.

**Negative examples for robustness**   Our graph-based recurrent retriever needs to be trained to discriminate between relevant and irrelevant paragraphs at each step. We therefore use negative examples along with the ground-truth paragraphs; to be more specific, we use two types of negative examples: (1) TF-IDF-based and (2) hyperlink-based ones. For single-hop QA, we only use the type (1). For multi-hop QA, we use both types, and the type (2) is especially important to prevent our retriever from being distracted by reasoning paths without correct answer spans. We typically set the number of the negative examples to 50.

---

[2]Appendix A.2 discusses the motivation, and Appendix C.4 shows results with an alternative approach.

**Loss function** For the sequential prediction task, we estimate $P(p_i|h_t)$ independently in Equation (3) and use the binary cross-entropy loss to maximize probability values of all the possible paths. Note that using the widely-used cross-entropy loss with the softmax normalization over $\mathbf{C}_t$ is not desirable here; maximizing the probabilities of $g$ and $g_r$ contradict with each other. More specifically, the loss function of $g$ at the $t$-th step is defined as follows:

$$L_{\mathrm{retr}}(p_t, h_t) = -\log P(p_t|h_t) - \sum_{\tilde{p} \in \tilde{\mathbf{C}}_t} \log(1 - P(\tilde{p}|h_t)), \tag{5}$$

where $\tilde{\mathbf{C}}_t$ is a set of the negative examples described above, and includes [EOE] for $t < |g|$. We exclude $p_r$ from $\tilde{\mathbf{C}}_1$ for the sake of our multi-path learning. The loss is also defined with respect to $g_r$ in the same way. All the model parameters, including those in BERT, are jointly optimized.

## 3.2 READING AND ANSWERING GIVEN REASONING PATHS

Our reader model first verifies each reasoning path in $\mathbf{E}$, and finally outputs an answer span $a$ from the most plausible reasoning path. This interplay is effective in making our framework robust; this is further discussed in Appendix A.3. We model the reader as a multi-task learning of (1) *reading comprehension,* that extracts an answer span from a reasoning path $E$ using a standard approach (Seo et al., 2017; Xiong et al., 2017; Devlin et al., 2019), and (2) *reasoning path re-ranking,* that re-ranks the retrieved reasoning paths by computing the probability that the path includes the answer.

For the reading comprehension task, we use BERT (Devlin et al., 2019), where the input is the concatenation of the question text and the text of all the paragraphs in $E$. This lets our reader to fully leverage the self-attention mechanism across the *concatenated* paragraphs in the retrieved reasoning paths; this paragraph interaction is crucial for multi-hop reasoning (Wang et al., 2019a).

We share the same model for re-ranking, and use the BERT's [CLS] representation to estimate the probability of selecting $E$ to answer the question:

$$P(E|q) = \sigma(w_n \cdot u_E) \quad \text{s.t.} \quad u_E = \mathrm{BERT}_{[\mathrm{CLS}]}(q, E) \in \mathbb{R}^D, \tag{6}$$

where $w_n \in \mathbb{R}^D$ is a weight vector. At the inference time, we select the best evidence $E_{best} \in \mathbf{E}$ by $P(E|q)$, and output the answer span by $S_{\mathrm{read}}$:

$$E_{best} = \underset{E \in \mathbf{E}}{\arg\max}\, P(E|q), \quad S_{\mathrm{read}} = \underset{i,j,\ i \le j}{\arg\max}\, P_i^{start} P_j^{end}, \tag{7}$$

where $P_i^{start}, P_j^{end}$ denote the probability that the $i$-th and $j$-th tokens in $E_{best}$ are the start and end positions, respectively, of the answer span, and are calculated by following Devlin et al. (2019).

**Training examples** To train the multi-task reader model, we use the ground-truth evidence paragraphs used for training our retriever. It is known to be effective in open-domain QA to use distantly supervised examples, which are not originally associated with the questions but include expected answer strings (Chen et al., 2017; Wang et al., 2018a; Hu et al., 2019). These distantly supervised examples are also effective to simulate the inference time process. Therefore, we combine distantly supervised examples from a TF-IDF retriever with the original supervised examples. Following the procedures in Chen et al. (2017), we add up to one distantly supervised example for each supervised example. We set the answer span as the string that matches $a$ and appears first.

To train our reader model to discriminate between relevant and irrelevant reasoning paths, we augment the original training data with additional negative examples to simulate incomplete evidence. In particular, we add paragraphs that appear to be relevant to the given question but actually do not contain the answer. For multi-hop QA, we select one ground-truth paragraph including the answer span, and swap it with one of the TF-IDF top ranked paragraphs. For single-hop QA, we simply replace the single ground-truth paragraph with TF-IDF-based negative examples which do not include the expected answer string. For the distorted evidence $\tilde{E}$, we aim at minimizing $P(\tilde{E}|q)$.

**Multi-task loss function** The objective is the sum of cross entropy losses for the span prediction and re-ranking tasks. The loss for the question $q$ and its evidence candidate $E$ is as follows:

$$L_{\mathrm{read}} = L_{\mathrm{span}} + L_{\mathrm{no\_answer}} = (-\log P_{y^{start}}^{start} - \log P_{y^{end}}^{end}) - \log P^r, \tag{8}$$

where $y^{start}$ and $y^{end}$ are the ground-truth start and end indices, respectively. $L_{\mathrm{no\_answer}}$ corresponds to the loss of the re-ranking model, to discriminate the distorted reasoning paths with no answers. $P^r$ is $P(E|q)$ if $E$ is the ground-truth evidence; otherwise $P^r = 1 - P(E|q)$. We mask the span losses for negative examples, in order to avoid unexpected effects to the span predictions.

## 4 EXPERIMENTS

### 4.1 EXPERIMENTAL SETUP

We evaluate our method in three open-domain Wikipedia-sourced datasets: HotpotQA, SQuAD Open and Natural Questions Open. We target all the English Wikipedia paragraphs for SQuAD Open and Natural Questions Open, and the first paragraph (introductory paragraph) of each article for HotpotQA following previous studies. More details can be found in Appendix B.

**HotpotQA** HotpotQA (Yang et al., 2018) is a human-annotated large-scale multi-hop QA dataset. Each answer can be extracted from a collection of 10 paragraphs in the *distractor* setting, and from the entire Wikipedia in the *full wiki* setting. Two evidence paragraphs are associated with each question for training. Our primary target is the full wiki setting due to its open-domain scenario, and we use the distractor setting to evaluate how well our method works in a closed scenario where the two evidence paragraphs are always included. The dataset also provides annotations to evaluate the prediction of supporting sentences, and we adapt our retriever to the supporting fact prediction. Note that this subtask is specific to HotpotQA. More details are described in Appendix A.5.

**SQuAD Open** SQuAD Open (Chen et al., 2017) is composed of questions from the original SQuAD dataset (Rajpurkar et al., 2016). This is a single-hop QA task, and a single paragraph is associated with each question in the training data.

**Natural Questions Open** Natural Questions Open (Lee et al., 2019) is composed of questions from the Natural Questions dataset (Kwiatkowski et al., 2019),[3] which is based on Google Search queries independently from the existing articles. A single paragraph is associated with each question, but our preliminary analysis showed that some questions benefit from multi-hop reasoning.

**Metrics** We report standard *F1* and *EM* scores for HotpotQA and SQuAD Open, and EM score for Natural Questions Open to evaluate the overall QA accuracy to find the correct answers. For HotpotQA, we also report *Supporting Fact F1 (SP F1)* and *Supporting Fact EM (SP EM)* to evaluate the sentence-level supporting fact retrieval accuracy. To evaluate the paragraph-level retrieval accuracy for the multi-hop reasoning, we use the following metrics: *Answer Recall (AR)*, which evaluates the recall of the answer string among top paragraphs (Wang et al., 2018a; Das et al., 2019), *Paragraph Recall (PR)*, which evaluates if at least one of the ground-truth paragraphs is included among the retrieved paragraphs, and *Paragraph Exact Match (P EM)*, which evaluates if both of the ground-truth paragraphs for multi-hop reasoning are included among the retrieved paragraphs.

**Evidence Corpus and the Wikipedia graph** We use English Wikipedia as the evidence corpus and do not use other data such as Google search snippets or external structured knowledge bases. We use the several versions of Wikipedia dumps for the three datasets (See Appendix B.5). To construct the Wikipedia graph, the hyperlinks are automatically extracted from the raw HTML source files. Directed edges are added between a paragraph $p_i$ and all of the paragraphs included in the target article. The constructed graph consists of 32.7M nodes and 205.4M edges. For HotpotQA we only use the introductory paragraphs in the graph that includes about 5.2M nodes and 23.4M edges.

**Implementation details** We use the pre-trained BERT models (Devlin et al., 2019) using the uncased base configuration ($d = 768$) for our retriever and the whole word masking uncased large (wwm) configuration ($d = 1024$) for our readers. We follow Chen et al. (2017) for the TF-IDF-based retrieval model and use the same hyper-parameters. We tuned the most important hyper-parameters, $F$, the number of the initial TF-IDF-based paragraphs, and $B$, the beam size, by mainly using the HotpotQA development set (the effects of increasing $F$ are shown in Figure 5 in Appendix C.3 along with the results with $B = 1$). If not specified, we set $B = 8$ for all the datasets, $F = 500$ for HotpotQA full wiki and SQuAD Open, and $F = 100$ for Natural Questions Open.

---

[3]We use train/dev/test splits provided by Min et al. (2019a), which can be downloaded from `https://drive.google.com/file/d/1qsN5Oyi_OtT2LyaFZFH26vT8Sqjb89-s/view`.

| | full wiki | | | | distractor | | | |
| | QA | | SP | | QA | | SP | |
| Models | F1 | EM | F1 | EM | F1 | EM | F1 | EM |
|---|---|---|---|---|---|---|---|---|
| Semantic Retrieval (Nie et al., 2019) | 58.8 | 46.5 | 71.5 | 39.9 | – | – | – | – |
| GoldEn Retriever (Qi et al., 2019) | 49.8 | – | 64.6 | – | – | – | – | – |
| Cognitive Graph (Ding et al., 2019) | 49.4 | 37.6 | 58.5 | 23.1 | – | – | – | – |
| DecompRC (Min et al., 2019c) | 43.3 | – | – | – | 70.6 | – | – | – |
| MUPPET (Feldman & El-Yaniv, 2019) | 40.4 | 31.1 | 47.7 | 17.0 | – | – | – | – |
| DFGN (Xiao et al., 2019) | – | – | – | – | 69.2 | 55.4 | – | – |
| QFE (Nishida et al., 2019) | – | – | – | – | 68.7 | 53.7 | 84.7 | **58.8** |
| Baseline (Yang et al., 2018) | 34.4 | 24.7 | 41.0 | 5.3 | 58.3 | 44.4 | 66.7 | 22.0 |
| Transformer-XH (Zhao et al., 2020) | 62.4 | 50.2 | 71.6 | 42.2 | – | – | – | – |
| Ours (Reader: BERT wwm) | **73.3** | **60.5** | **76.1** | **49.3** | **81.2** | **68.0** | **85.2** | 58.6 |
| Ours (Reader: BERT base) | 65.8 | 52.7 | 75.0 | 47.9 | 73.3 | 59.4 | 84.6 | 57.4 |

Table 1: **HotpotQA development set results**: QA and SP (supporting fact prediction) results on HotpotQA's full wiki and distractor settings. "–" denotes no results are available.

## 4.2 OVERALL RESULTS

Table 1 compares our method with previous published methods on the HotpotQA development set. Our method significantly outperforms all the previous results across the evaluation metrics under both the full wiki and distractor settings. Notably, our method achieves 14.5 F1 and 14.0 EM gains compared to state-of-the-art Semantic Retrieval (Nie et al., 2019) and 10.9 F1 gains over the concurrent Transformer-XH model (Zhao et al., 2020) on full wiki. We can see that our method, even with the BERT base configuration for our reader, significantly outperforms all the previous QA scores. Moreover, our method shows significant improvement in predicting supporting facts in the full wiki setting. We compare the performance of our approach to other models on the HotpotQA full wiki official hidden test set in Table 2. We outperform all the published and unpublished models including up-to-date work (marked with ♣) by large margins in terms of QA performance.

On SQuAD Open, our model outperforms the concurrent state-of-the-art model (Wang et al., 2019b) by 2.9 F1 and 3.5 EM scores as shown in Table 3. Due to the fewer lexical overlap between questions and paragraphs on Natural Questions, pipelined approaches using term-based retrievers often face difficulties finding associated articles. Nevertheless, our approach matches the performance of the best end-to-end retriever (ORQA), as shown in Table 4. In addition to its competitive performance, our retriever can be handled on a single GPU machine, while a fully end-to-end retriever in general requires industry-scale computational resources for training (Seo et al., 2019). More results on these two datasets are discussed in Appendix D.

## 4.3 PERFORMANCE OF REASONING PATH RETRIEVAL

We compare our retriever with competitive retrieval methods for HotpotQA full wiki, with $F = 20$.

**TF-IDF** (Chen et al., 2017), the widely used retrieval method that scores paragraphs according to the TF-IDF scores of the question-paragraph pairs. We simply select the top-2 paragraphs.
**Re-rank** (Nogueira & Cho, 2019) that learns to retrieve paragraphs by fine-tuning BERT to re-rank the top $F$ TF-IDF paragraphs. We select the top-2 paragraphs after re-ranking.
**Re-rank 2hop** which extends Re-rank to accommodate two-hop reasoning. It first adds paragraphs linked from the top TF-IDF paragraphs. It then uses the same BERT model to select the paragraphs.
**Entity-centric IR** is our re-implementation of Godbole et al. (2019) that is related to Re-rank 2hop, but instead of simply selecting the top two paragraphs, they re-rank the possible combinations of the paragraphs that are linked to each other.
**Cognitive Graph** (Ding et al., 2019) that uses the provided prediction results of the Cognitive Graph model on the HotpotQA development dataset.
**Semantic Retrieval** (Nie et al., 2019) that uses the provided prediction results of the state-of-the-art Semantic Retrieval model on the HotpotQA development dataset.

**Retrieval results** Table 5 shows that our recurrent retriever yields 8.8 P EM and 9.1 AR, leading to the improvement of 10.3 QA EM over Semantic Retrieval. The significant improvement from

| Models | QA | | SP | |
|---|---|---|---|---|
| (*: anonymous) | F1 | EM | F1 | EM |
| Semantic Retrieval | 57.3 | 45.3 | 70.8 | 38.7 |
| GoldEn Retriever | 48.6 | 37.9 | 64.2 | 30.7 |
| Cognitive Graph | 48.9 | 37.1 | 57.7 | 22.8 |
| Entity-centric IR | 46.3 | 35.4 | 43.2 | 0.06 |
| MUPPET | 40.3 | 30.6 | 47.3 | 16.7 |
| DecompRC | 40.7 | 30.0 | – | – |
| QFE | 38.1 | 28.7 | 44.4 | 14.2 |
| Baseline | 32.9 | 24.0 | 37.7 | 3.9 |
| HGN*♣ | 69.2 | 56.7 | **76.4** | **50.0** |
| MIR+EPS+BERT*♣ | 64.8 | 52.9 | 72.0 | 42.8 |
| Transformer-XH* | 60.8 | 49.0 | 70.0 | 41.7 |
| Ours | **73.0** | **60.0** | **76.4** | 49.1 |

Table 2: **HotpotQA full wiki test set results**: official leaderboard results (on November 6, 2019) on the hidden test set of the HotpotQA full wiki setting. Work marked with ♣ appeared after September 25.

| Models | F1 | EM |
|---|---|---|
| multi-passage (Wang et al., 2019b) | 60.9 | 53.0 |
| ORQA (Lee et al., 2019) | – | 20.2 |
| BM25+BERT (Lee et al., 2019) | – | 33.2 |
| Weaver (Raison et al., 2018) | – | 42.3 |
| RE$^3$ (Hu et al., 2019) | 50.2 | 41.9 |
| MUPPET (Feldman & El-Yaniv, 2019) | 46.2 | 39.3 |
| BERTserini (Yang et al., 2019) | 46.1 | 38.6 |
| DENSPI-hybrid (Seo et al., 2019) | 44.4 | 36.2 |
| MINIMAL (Min et al., 2018) | 42.5 | 34.7 |
| Multi-step Reasoner (Das et al., 2019) | 39.2 | 31.9 |
| Paragraph Ranker (Lee et al., 2018) | – | 30.2 |
| R$^3$ (Wang et al., 2018a) | 37.5 | 29.1 |
| DrQA (Chen et al., 2017) | – | 29.3 |
| Ours | **63.8** | **56.5** |

Table 3: **SQuAD Open results**: we report F1 and EM scores on the test set of SQuAD Open, following previous work.

| | | EM | |
|---|---|---|---|
| Models | | Dev | Test |
| ORQA (Lee et al., 2019) | | 31.3 | **33.3** |
| Hard EM (Min et al., 2019a) | | 28.8 | 28.1 |
| BERT + BM 25 (Lee et al., 2019) | | 24.8 | 26.5 |
| Ours | | **31.7** | 32.6 |

Table 4: **Natural Questions Open results**: we report EM scores on the test and development sets of Natural Questions Open, following previous work.

| Models | AR | PR | P EM | EM |
|---|---|---|---|---|
| Ours ($F = 20$) | 87.0 | 93.3 | 72.7 | 56.8 |
| TF-IDF | 39.7 | 66.9 | 10.0 | 18.2 |
| Re-rank | 55.1 | 85.9 | 29.6 | 35.7 |
| Re-rank 2hop | 56.0 | 70.1 | 26.1 | 38.8 |
| Entity-centric IR | 63.4 | 87.3 | 34.9 | 42.0 |
| Cognitive Graph | 76.0 | 87.6 | 57.8 | 37.6 |
| Semantic Retrieval | 77.9 | 93.2 | 63.9 | 46.5 |

Table 5: **Retrieval evaluation**: Comparing our retrieval method with other methods across Answer Recall, Paragraph Recall, Paragraph EM, and QA EM metrics.

Re-rank2hop to Entity-centric IR demonstrates that exploring entity links from the initially retrieved documents helps to retrieve the paragraphs with fewer lexical overlaps. On the other hand, comparing our retriever with Entity-centric IR and Semantic Retrieval shows the importance of learning to sequentially retrieve reasoning paths in the Wikipedia graph. It should be noted that our method with $F = 20$ outperforms all the QA EM scores in Table 1.

## 4.4 ANALYSIS

We conduct detailed analysis of our framework on the HotpotQA full wiki development set.

**Ablation study of our framework** To study the effectiveness of our modeling choices, we compare the performance of variants of our framework. We ablate the retriever with 1) *No recurrent module*, which removes the recurrence from our retriever, and computes the probability of each paragraph to be included in reasoning paths independently and selects the path with the highest joint probability path on the graph; 2) *No beam search*, which uses a greedy search ($B = 1$) in our recurrent retriever; 3) *No link-based negative examples,* which trains the retriever model without adding hyperlink-based negative examples besides TF-IDF-based negative examples.

We ablate the reader model with 1) *No reasoning path re-ranking*, which outputs the answer only with the best reasoning path from the retriever model, and 2) *No negative examples*, which trains the model only with the gold paragraphs, removing $L_{\text{no\_answer}}$ from $L_{\text{read}}$. During inference, "No negative examples" reads all the paths and outputs an answer with the highest answer probability.

| Settings ($F = 100$) | F1 | EM |
|---|---|---|
| full | 72.4 | 59.5 |
| retriever, no recurrent module | 52.5 | 42.1 |
| retriever, no beam search | 68.7 | 56.2 |
| retriever, no link-based negatives | 64.1 | 52.6 |
| reader, no reasoning path re-ranking | 70.1 | 57.4 |
| reader, no negative examples | 53.7 | 43.3 |

Table 6: **Ablation study**: evaluating different variants of our model on HotpotQA full wiki.

| Settings ($F = 100$) | F1 | EM |
|---|---|---|
| with hyperlinks | 72.4 | 59.5 |
| with entity linking system | 70.1 | 57.3 |

Table 7: **Performance with different link structures**: comparing our results on the Hotpot QA full wiki development set when we use an off-the-shelf entity linking system instead of the Wikipedia hyperlinks.

| Settings ($F = 100$) | | F1 | EM |
|---|---|---|---|
| Adaptive retrieval | | 72.4 | 59.5 |
| $L$-step retrieval | $L = 1$ | 45.8 | 35.5 |
| | $L = 2$ | 71.4 | 58.5 |
| | $L = 3$ | 70.1 | 57.7 |
| | $L = 4$ | 66.3 | 53.9 |

Table 8: **Performance with different reasoning path length**: comparing the performance with different path length on HotpotQA full wiki. $L$-step retrieval sets the number of the reasoning steps to a fixed number.

| ($F = 100$) | Retriever | Reader | EM |
|---|---|---|---|
| Avg. # of $L$ | 1.96 | 2.21 | with $L$ |
| 1 | 539 | 403 | 31.2 |
| 2 | 6,639 | 5,655 | 60.0 |
| 3 | 227 | 1,347 | 63.0 |

Table 9: **Statistics of the reasoning paths**: the average length and the distribution of length of the reasoning paths selected by our retriever and reader for HotpotQA full wiki. Avg. EM represents QA EM performance.

**Ablation results**  Table 6 shows that removing any of the listed components gives notable performance drop. The most critical component in our retriever model is the recurrent module, dropping the EM by 17.4 points. As shown in Figure 1, multi-step retrieval often relies on information mentioned in another paragraph. Therefore, without conditioning on the previous time steps, the model fails to retrieve the complete evidence. Training without hyperlink-based negative examples results in the second largest performance drop, indicating that the model can be easily distracted by reasoning paths without a correct answer and the importance of negative sampling for training. Replacing the beam search with the greedy search gives a performance drop of about 4 points on EM, which demonstrates that being aware of the graph structure is helpful in finding the best reasoning paths.

Performance drop by removing the reasoning path re-ranking indicates the importance of verifying the reasoning paths in our reader. Not using negative examples to train the reader degrades EM more than 16 points, due to the over-confident predictions as discussed in Clark & Gardner (2018).

**The performance with an off-the-shelf entity linking system**  Although the existence of the hyperlinks is not special on the web, one question is how well our method works without the Wikipedia hyperlinks. We evaluate our method on the development set of HotpotQA full wiki with an off-the-shelf entity linking system (Ferragina & Scaiella, 2011) to construct the document graph in our method. More details about this experimental setup can be found in Appendix B.7. Table 7 shows that our approach with the entity linking system shows only 2.3 F1 and 2.2 EM lower scores than those with the hyperlinks, still achieving the state of the art. This suggests that our approach is not restricted to the existence of the hyperlink information, and using hyperlinks is promising.

**The effectiveness of arbitrary-step retrieval**  The existing iterative retrieval methods fix the number of reasoning steps (Qi et al., 2019; Das et al., 2019; Godbole et al., 2019; Feldman & El-Yaniv, 2019), while our approach accommodates arbitrary steps of reasoning. We also evaluate our method by fixing the length of the reasoning path ($L = \{1, 2, 3, 4\}$). Table 8 shows that out adaptive retrieval performs the best, although the length of all the annotated reasoning paths in HotpotQA is two. As discussed in Min et al. (2019b), we also observe that some questions are answerable based on a single paragraph, where our model flexibly selects a single paragraph and then terminates retrieval.

**The effectiveness of the interplay between retriever and reader**  Table 6 shows that the interplay between our retriever and reader models is effective. To understand this, we investigate the length of reasoning paths selected by our retriever and reader, and their final QA performance. Table 9 shows that the average length selected by our reader is notably longer than that by our retriever. Table 9 also presents the EM scores averaged over the questions with certain length of reasoning paths

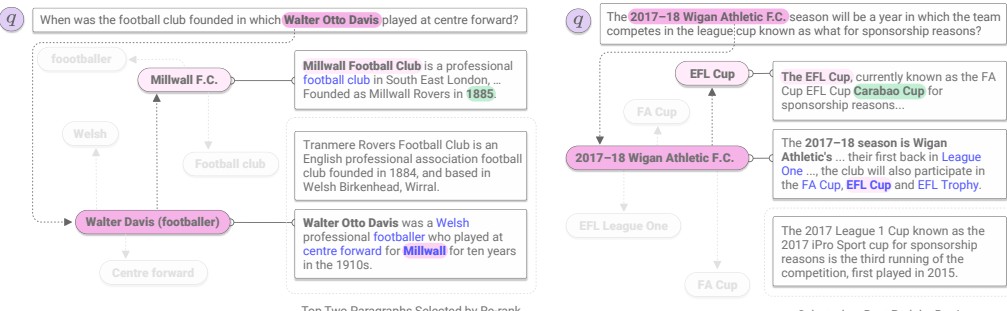

Figure 3: Reasoning examples by our model (two paragraphs connected by a dotted line) and Re-rank (the bottom two paragraphs). Highlighted text denotes a bridge entity, and blue-underlined text represents hyperlinks.

Figure 4: Reasoning examples by our retriever (the bottom paragraph) and our reader (two paragraphs connected by a dotted line). Highlighted text denotes a bridge entity, and blue-underlined text represents hyperlinks.

($L = \{1, 2, 3\}$). We observe that our framework performs the best when it selects the reasoning paths with $L = 3$, showing 63.0 EM score. Based on these observations, we expect the retriever favors a shorter path, while the reader tends to select a longer and more convincing multi-hop reasoning path to derive an answer string.

**Qualitative examples of retrieved reasoning paths** Finally, we show two examples from HotpotQA full wiki, and Appendix C.5 presents more qualitative examples. In Figure 3, our approach successfully retrieves the correct reasoning path and answers correctly, while Re-rank fails. The top two paragraphs next to the graph are the introductory paragraphs of the two entities on the reasoning path, and the paragraph at the bottom shows the wrong paragraph selected by Re-rank. The "Millwall F.C." has fewer lexical overlaps and the bridge entity "Millwall" is not stated in the given question. Thus, Re-rank chooses a wrong paragraph with high lexical overlaps to the given question. In Figure 4, we compare the reasoning paths ranked highest by our retriever and reader. Although the gold path is included among the top 8 paths selected by the beam search, our retriever model selects a wrong paragraph as the best reasoning path. By re-ranking the reasoning paths, the reader eventually selects the correct reasoning path ("2017-18 Wigan Athletic F.C. season" → "EFL Cup"). This example shows the effectiveness of the strong interplay of our retriever and reader.

## 5 CONCLUSION

This paper introduces a new graph-based recurrent retrieval approach, which retrieves reasoning paths over the Wikipedia graph to answer multi-hop open-domain questions. Our retriever model learns to sequentially retrieve evidence paragraphs to form the reasoning path. Subsequently, our reader model re-ranks the reasoning paths, and it determines the final answer as the one extracted from the best reasoning path. Our experimental results significantly advance the state of the art on HotpotQA by more than 14 points absolute gain on the full wiki setting. Our approach also achieves the state-of-the-art performance on SQuAD Open and Natural Questions Open without any architectural changes, demonstrating the robustness of our method. Our method provides insights into the underlying entity relationships, and the discrete reasoning paths are helpful in interpreting our framework's reasoning process. Future work involves end-to-end training of our graph-based recurrent retriever and reader for improving upon our current two-stage training.

ACKNOWLEDGMENTS

We acknowledge grants from ONR N00014-18-1-2826, DARPA N66001-19-2-403, NSF (IIS1616112, IIS1252835), and Samsung GRO. We thank Sewon Min, David Wadden, Yizhong Wang, Akhilesh Gotmare, Tong Niu, and UW NLP group and Salesforce research members for their insightful discussions. We would also like to show our gratitude to Melvin Gruesbeck for providing us with the artistic figures presented in this paper. We thank the anonymous reviewers for their helpful and thoughtful comments. Akari Asai is supported by The Nakajima Foundation Fellowship.

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

## APPENDIX

## A  DETAILS ABOUT MODELING

### A.1  A NORMALIZED RNN

We decompose Equation (4) as follows:

$$a_{t+1} = W_r[h_t; w_i] + b_r, \quad h_{t+1} = \frac{\alpha}{\|a_{t+1}\|} a_{t+1}, \tag{9}$$

where $W_r \in \mathbb{R}^{d \times 2d}$ is a weight matrix, $b_r \in \mathbb{R}^d$ is a bias vector, and $\alpha \in \mathbb{R}^1$ is a scalar parameter (initialized with 1.0). We set the global initial state $a_1$ to a parameterized vector $s \in \mathbb{R}^d$, and we also parameterize an [EOE] vector $w_{[\text{EOE}]} \in \mathbb{R}^d$ for the [EOE] symbol. The use of $w_i$ for both the input and output layers is inspired by Inan et al. (2017); Press & Wolf (2017). In addition, we align the norm of $w_{[\text{EOE}]}$ with those of $w_i$, by applying layer normalization (Ba et al., 2016) of the last layer in BERT because $w_{[\text{EOE}]}$ is used along with the BERT outputs. Without the layer normalization, the $L2$-norms of $w_i$ and $w_{[\text{EOE}]}$ can be quite different, and the model can easily discriminate between them by the difference of the norms.

### A.2  QUESTION-PARAGRAPH ENCODING IN OUR RETRIEVER COMPONENT

Equation (2) shows that we compute each paragraph representation $w_i$ conditioned on the question $q$. An alternative approach is separately encoding the paragraphs and the question, to directly retrieve paragraphs (Lee et al., 2019; Seo et al., 2019; Das et al., 2019). However, due to the lack of explicit interactions between the paragraphs and the question, such a neural retriever using question-independent paragraph encodings suffers from compressing the necessary information into fixed-dimensional vectors, resulting in low performance on entity-centric questions (Lee et al., 2019). It has been shown that attention-based paragraph-question interactions improve the retrieval accuracy if the retrieval scale is tractable (Wang et al., 2018a; Lee et al., 2018). There is a trade-off between the scalability and the accuracy, and this work aims at striking the balance by jointly using the lexical matching retrieval and the graphs, followed by the rich question-paragraph encodings.

**A question-independent variant**  We can also formulate our retriever model by using a question-independent approach. There are only two simple modifications. First, we reformulate Equation (2) as follows:

$$w_i = \text{BERT}_{[\text{CLS}]}(p_i), \tag{10}$$

where we no longer input the question $q$ together with the paragraphs. Next, we condition the initial RNN state $h_1$ on the question information. More specifically, we compute $h_1$ by using Equation (4) as follows:

$$w_q = \text{BERT}_{[\text{CLS}]}(q), \tag{11}$$

$$h_1 = \text{RNN}(h_1', w_q), \tag{12}$$

where $w_q$ is computed by using the same BERT encoder as in Equation (10), and $h_1'$ is the original $h_1$ used in our question-dependent approach as described in Appendix A.1. The remaining parts are exactly the same, and we can perform the reasoning path retrieval in the same manner.

### A.3 WHY IS THE INTERPLAY IMPORTANT?

Our retriever model learns to predict plausibility of the reasoning paths by capturing the paragraph interactions through the BERT's [CLS] representations, after *independently* encoding the paragraphs along with the question; this makes our retriever scalable to the open-domain scenario. By contrast, our reader jointly learns to predict the plausibility and answer the question, and moreover, fully leverages the self-attention mechanism across the *concatenated* paragraphs in the retrieved reasoning paths; this paragraph interaction is crucial for multi-hop reasoning (Wang et al., 2019a). In summary, our retriever is scalable, but the top-1 prediction is not always enough to fully capture multi-hop reasoning to answer the question. Therefore, the additional re-ranking process mitigates the uncertainty and makes our framework more robust.

### A.4 HANDLING YES-NO QUESTIONS IN OUR READER COMPONENT

In the HotpotQA dataset, we need to handle yes-no questions as well as extracting answer spans from the paragraphs. We treat the two special types of the answers, yes and no, by extending the re-ranking model in Equation (6). In particular, we extend the binary classification to a multi-class classification task, where the positive "answerable" class is decomposed into the following three classes: span, yes, and no. If the probability of "yes" or "no" is the largest among the three classes, our reader directly outputs the label as the answer, without any span extraction. Otherwise, our reader uses the span extraction model to output the answer.

### A.5 SUPPORTING FACT PREDICTION IN HOTPOTQA

We adapt our recurrent retriever to the subtask of the supporting fact prediction in HotpotQA (Yang et al., 2018). The task is outputting sentences which support to answer the question. Such supporting sentences are annotated for the two ground-truth paragraphs in the training data. Since our framework outputs the most plausible reasoning path $E$ along with the answer, we can add an additional step to select supporting facts (sentences) from the paragraphs in $E$. We train our recurrent retriever by using the training examples for the supporting fact prediction task, where the model parameters are not shared with those of our paragraph retriever. We replace the question-paragraph encoding in Equation (2) with question-answer-sentence encoding for the task, where a question string is concatenated with its answer string. The answer string is the ground-truth one during the training time. We then maximize the probability of selecting the ground-truth sequence of the supporting fact sentences, while setting the other sentences as negative examples. At test time, we use the best reasoning path and its predicted answer string from our retriever and reader models to finally output the supporting facts for each question. The supporting fact prediction task is performed after finalizing the reasoning path and the answer for each question, and hence this additional task does not affect the QA accuracy.

## B DETAILS ABOUT EXPERIMENTS

### B.1 DATASET DETAILS OF HOTPOTQA, SQUAD OPEN AND NATURAL QUESTIONS OPEN

**HotpotQA**  The HotpotQA training, development, and test datasets contain 90,564, 7,405 and 7,405 questions, respectively. To train our retriever model for the distractor setting, we use the distractor training data, where only the original ten paragraphs are associated with each question. The retriever model trained with this setting is also used in our ablation study as "retriever, no link-based negatives" in Table 6. For the full wiki setting, we train our retriever model with the data augmentation technique and the additional negative examples described in Section 3.1.2. We use the same reader model, for both the settings, trained with the augmented additional references and the negative examples described in Section 3.2.

**SQuAD Open and Natural Questions Open**  For SQuAD Open, we use the original training set (containing 78,713 questions) as our training data, and the original development set (containing 10,570 questions) as our test data. For Natural Questions Open, we follow the dataset splits provided by Min et al. (2019a), and the training, development and test datasets contain 79,168, 8,757 and 3,610, respectively. For both the SQuAD Open and Natural Questions Open, we train our reader on

the original examples with the augmented additional negative examples and the distantly supervised examples described in Section 3.2.

## B.2 DERIVING GROUND-TRUTH REASONING PATHS

Section 3.1.2 describes our training strategy for our recurrent retriever. We apply the data augmentation technique to HotpotQA and Natural Questions to consider multi-hop reasoning. To derive the ground-truth reasoning path $g$, we use the ground-truth evidence paragraphs associated with the questions in the training data for each dataset. For SQuAD and Natural Questions Open, each training example has only single paragraph $p$, and thus it is trivial to derive $g$ as $[p, [EOE]]$. For the multi-hop case, HotpotQA, we have two ground-truth paragraphs $p_1, p_2$ for each question. Assuming that $p_2$ includes the answer string, we set $g = [p_1, p_2, [EOE]]$.

## B.3 DETAILS ABOUT NEGATIVE EXAMPLES FOR OUR READER MODEL IN SQUAD OPEN AND NATURAL QUESTIONS OPEN

To train our reader model for SQuAD Open, in addition to the TF-IDF top-ranked paragraphs, we add two types of additional negative examples: (i) paragraphs, which do not include the answer string, from the originally annotated articles, and (ii) "unanswerable" questions from SQuAD 2.0 (Rajpurkar et al., 2018). For Natural Questions Open, we add negative examples of the type (i).

## B.4 TRAINING SETTINGS

To use the pre-trained BERT models, we used the public code base, pytorch-transformers,[4] written in PyTorch.[5] For optimization, we used the code base's implementation of the Adam optimizer (Kingma & Ba, 2015), with a weight-decay coefficient of $0.01$ for non-bias parameters. A warm-up strategy in the code base was also used, with a warm-up rate of $0.1$. Most of the settings follow the default settings. To train our recurrent retriever, we set the learning rate to $3 \cdot 10^{-5}$, and the maximum number of the training epochs to three. The mini-batch size is four; a mini-batch example consists of a question with its corresponding paragraphs. To train our reader model, we set the learning rate to $3 \cdot 10^{-5}$, and the maximum number of training epochs to two. Empirically we observe better performance with a larger batch size as discussed in previous work (Liu et al., 2019; Ott et al., 2018), and thus we set the mini-batch size to 120. A mini-batch example consists of a question with its evidence paragraphs. We will release our code to follow our experiments.

## B.5 THE WIKIPEDIA DUMPS FOR EACH DATASET

For HotpotQA full wiki, we use the pre-processed English Wikipedia dump from October, 2017, provided by the HotpotQA authors.[6] For Natural Questions Open, we use the English Wikipedia dump from December 20, 2018, following Lee et al. (2019) and Min et al. (2019a). For SQuAD Open, we use the Wikipedia dump provided by Chen et al. (2017).

Although using a single dump for different open-domain QA datasets is a common practice (Chen et al., 2017; Wang et al., 2018a; Lee et al., 2018), this potentially causes inconsistent or even unfair evaluation across different experimental settings, due to the temporal inconsistency of the Wikipedia articles. More concretely, every Wikipedia article is editable and and as a result, a fact can be rephrased or could be removed. For instance, a question from the SQuAD development set, "Where does Kenya rank on the CPI scale?" is originally paired with a paragraph from the article of Kenya. Based on a single sentence "Kenya ranks low on Transparency International's Corruption Perception Index (CPI)" from the paragraph, an annotated answer span is "low." However, this sentence has been rewritten as "Kenya has a high degree of corruption according to Transparency International's Corruption Perception Index (CPI)" in a later version of the same article.[7] This is problematic considering the major evaluation metrics based on string matching.

---

[4]`https://github.com/huggingface/pytorch-transformers`.
[5]`https://pytorch.org/`.
[6]`https://hotpotqa.github.io/wiki-readme.html`.
[7]`https://en.wikipedia.org/wiki/Kenya` on October 25, 2019

Another problem exists especially in Natural Questions Open. The dataset contains real Google search queries, and some of them reflect temporal trends at the time when the queries were executed. If a query is related to a TV show broadcasted in 2018, we can hardly expect to extract the answer from a dump in 2017.

Like this, although Wikipedia is a useful knowledge source for open-domain QA research, its rapidly evolving nature should be considered more carefully for the reproducibility. We will make all of the data including pre-processed Wikipedia articles for each experiment available for future research.

### B.6 Details about Initial Candidates $C_1$ selection

To retrieve the initial candidates $C_1$ for each question, we use a TF-IDF based retriever with the bi-gram hashing (Chen et al., 2017). For HotpotQA full wiki, we retrieve top $F$ introductory paragraphs, for each question, from a corpus including all the introductory paragraphs. For SQuAD Open and Natural Questions Open, we first retrieve 50 Wikipedia articles through the same TF-IDF retriever, and further run another TF-IDF-based paragraph retriever (Clark & Gardner, 2018; Min et al., 2019a) to retrieve $F$ paragraphs in total.

### B.7 Details about entity linking experiment

We experiment with a variant of our approach, where we incorporate an entity linking system with our framework, in place of the Wikipedia hyperlinks. In this experiment, we first retrieve seed paragraphs using TF-IDF ($F = 100$), and run an off-the-shelf entity linker (TagMe by Ferragina & Scaiella (2011)) over the paragraphs. If the entity linker detects some entities, we retrieve their corresponding Wikipedia articles, and add edges from the seed paragraphs to the entity-linked paragraphs. Once we build the graph, then we re-run all of the experiments while the other components are exactly the same. We use the TagMe official Python wrapper.[8]

## C Additional Results on HotpotQA

### C.1 Upper-bound of our retrieval module

For scalability and computational efficiency, we bootstrap our retrieval module with TF-IDF retrieval; we first retrieval $F$ paragraphs using TF-IDF with the method described in Section B.6 and initialize $C_1$ with these TF-IDF paragraphs. Although we expand our candidate paragraphs at each time step using the Wikipedia graph, if our method failed to retrieve paragraphs a few-hops away from the answer paragraphs, it is likely to fail to reach the answer paragraphs. To estimate the paragraph EM upper-bound, we have checked if two gold paragraphs are included in the top 20 TF-IDF paragraphs and their hyperlinked paragraphs in the HotpotQA full wiki setting. We found that for 75.4% of the questions, all of the gold paragraphs are included in the collections of the TF-IDF paragraphs and the hyperlinked paragraphs. Also, it should be noted when we only consider the TF-IDF retrieval results, the upper-bound drops to 35.1%, which suggests that the TF-IDF-based retrieval cannot effectively discover the paragraphs multi-hop away due to the few lexical overlap. When we increase the number of $F$ to 100 and 500, the upper-bound reaches 84.1% and 89.2%, respectively.

### C.2 Per-Category Question Answering and Retrieval Performance on HotpotQA full wiki

In HotpotQA, there are two types of questions, *bridge* and *comparison*. While comparison-type questions explicitly mention the two entities related to the given questions, in bridge-type questions, the bridge entities are rarely explicitly stated. This makes it hard for a retrieval system to discover the paragraphs entailed by the bridge entities only.

We evaluate the question answering and paragraph retrieval performance for each of the two question types. We compare the PR, P EM and QA EM for each of the two categories with two state-of-the-art models, Cognitive Graph (Ding et al., 2019) and Semantic Retrieval (Nie et al., 2019). Here, we set our initial TF-IDF number $F$ to 500. Table 10 shows that our retriever yields 16.5 P EM

---

[8] https://github.com/marcocor/tagme-python

| Models | Total (7,405) | | | Bridge (5,918) | | | Comp (1,487) | | |
|---|---|---|---|---|---|---|---|---|---|
| | PR | P EM | EM | PR | P EM | EM | PR | P EM | EM |
| Ours | **94.3** | **75.7** | **60.5** | **93.9** | **73.7** | **57.8** | 98.7 | 83.5 | **70.5** |
| Cognitive Graph (Ding et al., 2019) | 87.6 | 57.8 | 37.5 | 84.8 | 51.8 | 36.1 | 98.6 | 81.6 | 53.7 |
| Semantic Retrieval (Nie et al., 2019) | 93.2 | 63.9 | 46.5 | 91.6 | 57.2 | 42.7 | **99.7** | **90.6** | 61.7 |

Table 10: **Retrieval evaluation**: Comparing our retrieval method with other methods across Answer Recall, Paragraph Recall, Paragraph EM, and QA EM metrics.

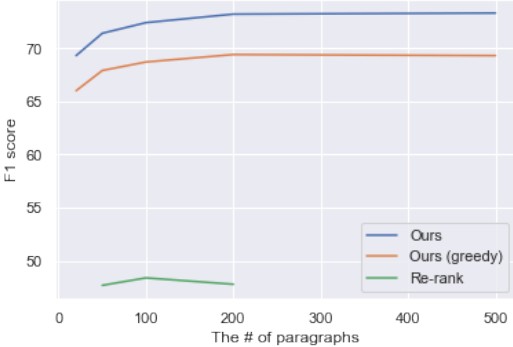

Figure 5: **Robustness to the increase of** $F$. We compare the F1 scores of our model, our model without a beam search and Re-rank with different number of $F$.

gain and 15.1 EM gain over Semantic Retrieval for the challenging bridge-type questions. For the comparison-type questions, our method achieves almost 10 point higher QA EM than Semantic Retrieval. We observed that some of the comparison-type questions can be answered based on single paragraph, and thus our model selects only one paragraph for some of these comparison-type questions, resulting in lower P EM scores on the comparison-type questions. We show several examples of the questions where we can answer based on single paragraph in Section C.5.

## C.3    ON THE ROBUSTNESS TO THE INCREASE OF THE PARAGRAPHS

As we discussed in 3.1.1, we aim at significantly reducing the search space and thus scaling the number of initial TF-IDF candidates. Increasing the number of the initial retrieved paragraphs often improves the recall of the evidence paragraphs of the datasets. On the other hand, increasing the candidate paragraphs introduces additional noises, may distract models, and eventually hurt the performance (Kratzwald & Feuerriegel, 2018). We compare the performance of three different approaches: (i) *ours*, (ii) *ours (greedy, without reasoning path re-ranking)*, and (iii) *Re-rank*. We increase the number of the TF-IDF-based retrieved paragraphs from 10 to 500 (For Re-rank, we compare the performance up to 200 paragraphs). Figure 5 clearly shows that our approach is robust towards the increase of the initial candidate paragraphs, and thus can constantly yield performance gains with more candidate paragraphs. Our approach with the greedy search also shows performance improvements; however, after a certain number, the greedy approach stops improving the performance. Re-rank starts suffering from the noises caused by many distracting paragraphs included in the initial candidate paragraphs at $F = 200$.

## C.4    RESULTS OF QUESTION-INDEPENDENT PARAGRAPH ENCODING FOR OUR RETRIEVER

To show the importance of the question-paragraph encoding in our retriever model, we conduct an experiment on the development set of HotpotQA, by replacing it with the question-independent encoding described in Appendix A.2. For a fair comparison, we use the same initial TF-IDF-based retrieval (only for the full wiki setting), hyperlink-based Wikipedia graph, beam search, and reader model (BERT wwm). We train the alternative model without using the data augmentation technique (described in Section 3.1.2) for quick experiments.

| Encoding method | full wiki ($F = 100$) | | distractor | |
|---|---|---|---|---|
| | QA F1 | QA EM | QA F1 | QA EM |
| Question-dependent (our main model) | 64.1 | 52.6 | 81.2 | 68.0 |
| Question-independent | 47.3 | 37.8 | 80.0 | 66.4 |

Table 11: **Effects of the question-dependent paragraph encoding**: Comparing our retriever model with and without the query-dependent encoding. For our question-dependent approach, the full wiki results correspond to "retriever, no link-based negatives" in Table 6, and the distractor results correspond to "Ours (Reader: BERT wwm)" Table 1, to make the results comparable.

Table 11 shows the results in both the full wiki and distractor settings. As seen in this table, the QA F1 and EM performance significantly deteriorates on the full wiki setting, which demonstrates the importance of the question-dependent encoding for complex and entity-centric open-domain question answering.

We can also see that the performance drop on the distractor setting is much smaller than that on the full wiki setting. This is due to its closed nature; for each question, we are given only ten paragraphs and the two gold paragraphs are always included, which significantly narrows the searching space down and makes the retrieval task much easier than that in the full wiki setting. Therefore, our recurrent retriever model is likely to discover the gold reasoning paths by the beam search, and our reader model can select the gold paths by the robust re-ranking approach. To verify this hypothesis, we checked the P EM score as a retrieval accuracy in the distractor setting. If we only consider the top-1 path from the beam search, the P EM score of the question-independent model is 12% lower than that of our question-dependent model. However, if we consider all the reasoning paths produced by the beam search, the coverage of the gold paths is almost the same. As a result, our reader model can perform similarly with both the question-dependent/independent approaches. This additionally shows the robustness of our re-ranking approach.

## C.5   More qualitative analysis on the reasoning path on HotpotQA full wiki

In this section, we conduct more qualitative analysis on the reasoning paths predicted by our model. Explicitly retrieving plausible reasoning paths and re-ranking the paths provide us interpretable insights into the underlying entity relationships used for multi-hop reasoning.

As shown in Table 9, our model flexibly selects one or more paragraphs for each question. To understand these behaviors, we conduct qualitative analysis on these examples whose reasoning paths are shorter or longer than the original gold reasoning paths.

**Reasoning path only with single paragraph**   First, we show two examples (one is a bridge-type question and the other is a comparison-type question), where our retriever selects single paragraph and terminates without selecting any additional paragraphs.

The bridge-type question in Table 12 shows that, while originally this question requires a system to read two paragraphs, **Before I Go to Sleep (film)** and **Nicole Kidman**, our retriever and reader eventually choose **Nicole Kidman** only. The second paragraph has a lot of lexical overlaps to the given question, and thus, a system may not need to read both of the paragraphs to answer.

The comparison-type question in Table 12 also shows that even comparison-type questions do not always require two paragraphs to answer the questions, and our model only selects one paragraph necessary to answer the given example question. In this example, the question has large lexical overlap with one of the ground-truth paragraph (**The Bears and I**), resulting in allowing our model to answer the question based on the single paragraph.

Min et al. (2019b) also observed that some of the questions do not necessarily require multi-hop reasoning, while HotpotQA is designed to require multi-hop reasoning (Yang et al., 2018). In that sense, we can say that our method automatical detects potentially single-hop questions.

**Reasoning path with three paragraphs**   All of the HotpotQA questions are authored by annotators who are shown two relevant paragraphs, and thus, originally the length of ground-truth reason-

---

**Q [bridge]**: Before I Go to Sleep stars an Australian actress, producer and occasional what?

---

**Before I Go to Sleep (film)**: Before I Go to Sleep is a 2014 mystery psychological thriller film written and directed by Rowan Joff and based on the 2011 novel of the same name by S. J. Watson. An international co-production between the United Kingdom, the United States, France, and Sweden, the film stars Nicole Kidman, Mark Strong, Colin Firth, and Anne-Marie Duff.

- - - - - - - - - - - - - - - - - - - - - - - - - - - - - - - - - - - - - - - - - -

**Nicole Kidman**: Nicole Mary Kidman, is an Australian actress, producer and occasional singer. She is the recipient of several awards, including an Academy Award, two Primetime Emmy Awards, a BAFTA Award, three Golden Globe Awards, and the Silver Bear for Best Actress.

- - - - - - - - - - - - - - - - - - - - - - - - - - - - - - - - - - - - - - - - - -

Annotated reasoning path **Before I Go to Sleep (film)** → **Nicole Kidman**
Predicted reasoning path: **Nicole Kidman**

---

**Q [comparison]**: In between The Bears and I and Oceans which was released on July 31, 1974, by Buena Vista Distribution?

---

**The Bears and I**: The Bears and I is a 1974 American drama film directed by Bernard McEveety and written by John Whedon. The film stars Patrick Wayne, Chief Dan George, Andrew Duggan, Michael Ansara and Robert Pine. *The film was released on July 31, 1974, by Buena Vista Distribution.*

- - - - - - - - - - - - - - - - - - - - - - - - - - - - - - - - - - - - - - - - - -

**Oceans (film)**: Oceans is a 2009 French nature documentary film directed, produced, co-written, and narrated by Jacques Perrin, with Jacques Cluzaud as co-director.

- - - - - - - - - - - - - - - - - - - - - - - - - - - - - - - - - - - - - - - - - -

Annotated reasoning path: **The Bears and I**, **Oceans (film)**
Predicted reasoning path: **The Bears and I**

---

Table 12: Two examples of the questions that our model retrieves a reasoning path with only one paragraph. We partly remove sentences irrelevant to the questions. Words in red correspond to the answer strings.

ing paths is always two. On the other hand, as our model accommodates arbitrary steps of reasoning, it often selects reasoning paths longer than the original annotations as shown in Table 9. *When our model selects a longer reasoning path for a HotpotQA question, does it contain paragraphs that provide additional evidence?* We show an example in Table 13, so as to answer this question. Our model selects an additional paragraph, **Blue Jeans (Lana Del Rey song)** at the first step, and then selects the two annotated gold paragraphs. This first paragraph is strongly relevant to the given question, but does not contain the answer. This additional evidence might help the reader to find the correct bridge entity ("Back to December").

## C.6 QUALITATIVE ANALYSIS ON THE REASONING PATH ON HOTPOTQA DISTRACTOR

Although the main focus in this paper is on open-domain QA, we show the state-of-the-art performance on the HotpotQA distractor setting as well with the exactly same architecture. We conduct qualitative analysis to understand our model's behavior in the closed setting. In this setting, the two ground-truth paragraphs are always given for each question.

Table 14 shows two examples from the HotpotQA distractor setting. In the first example, **P1** and **P2** are its corresponding ground-truth paragraphs. At the first time step, our retriever does not expect that **P2** is related to the evidence to answer the question, as the retriever is not aware of the bridge entity, "Pasek & Paul". If we simply adopt the Re-rank strategy, **P3** with the second highest probability is selected, resulting in a wrong paragraph selection. In our framework, our retriever is conditioned on the previous retrieval history and thus, at the second time step, it chooses the correct paragraph, **P2**, lowering the probability of **P3**. This clearly shows the effectiveness of our multi-step retrieval method in the closed setting as well. At the third step, our model stops the prediction by

**Q**: Yoann Lemoine, a French video director, has created music videos for Lana Del Rey, Katy Perry, and an orchestral country pop ballad by which top pop artist?

**Yoann Lemoine**: Yoann Lemoine (born 16 March 1983) is a French music video director, graphic designer and singer-songwriter. His most notable works include his music video direction for Katy Perry's "Teenage Dream", Taylor Swift's single "Back to December", Lana Del Rey's "Born to Die" and Mystery Jets' "Dreaming of Another World".

- - - - - - - - - - - - - - - - - - - - - - - - - - - - - - - - - - - - - - - - - -

**Back to December**: "Back to December" is a song written and recorded by American singer/songwriter Taylor Swift for her third studio album "Speak Now" (2010). "Back to December" is considered an orchestral country pop ballad and its lyrics are a remorseful plea for forgiveness for breaking up with a former lover.

- - - - - - - - - - - - - - - - - - - - - - - - - - - - - - - - - - - - - - - - - -

**Blue Jeans (Lana Del Rey song)**: "Blue Jeans" is a song by American singer-songwriter Lana Del Rey for her second studio album "Born to Die" (2012). Produced by Emile Haynie, the song was written by Del Rey, Haynie, and Dan Heath. Charting across Europe and Asia, "Blue Jeans" reached the top 10 in Belgium, Poland, and Israel. The second was shot and directed by Yoann Lemoine, featuring film noir elements and crocodiles.

- - - - - - - - - - - - - - - - - - - - - - - - - - - - - - - - - - - - - - - - - -

Annotated reasoning path: **Yoann Lemoin → Back to December**
Predicted reasoning path: **Blue Jeans (Lana Del Rey song) → Yoann Lemoin → Back to December**

Table 13: An example question where our model predicts reasoning paths of the length of three. Our model expects that the question is answerable based on the last paragraph of the annotated path.

**Q**: Which songwriting duo composed music for "La La Land", and created lyrics for "A Christmas Story: The Musica"?

| | | | |
|---|---|---|---|
| **P1**: A Christmas Story: The Musical is a musical version of the film "A Christmas Story ... The musical has music and lyrics written by Pasek & Paul and the book by Joseph Robinette. | 0.98 ✓ | 0.00 | 0.00 |
| **P2**: Benj Pasek and Justin Paul, known together as Pasek and Paul, are an American songwriting duo and composing team for musical theater, films, and television. ... they won both the Golden Globe and Academy Award for Best Original Song for the song "City of Stars". | 0.08 | 0.89 ✓ | 0.00 |
| **P3**: La La Land" is a song recorded by American singer Demi Lovato. It was written by Lovato, Joe Jonas, Nick Jonas and Kevin Jonas and produced by the Jonas Brothers alongside John Fields, for Lovato's debut studio album, "Don't Forget" (2008). | 0.12 | 0.00 | 0.00 |

**Q**: Alexander Kerensky was defeated and destroyed by the Bolsheviks in the course of a civil war that ended when ?

| | | | |
|---|---|---|---|
| **P1**: The Socialist Revolutionary Party, or Party of Socialists-Revolutionaries sery") was a major political party in early 20th century Russia and a key player in the Russian Revolution. ... The anti-Bolshevik faction of this party, known as the Right SRs, which remained loyal to the Provisional Government leader Alexander Kerensky was defeated and destroyed by the Bolsheviks in the course of the Russian Civil War and subsequent persecution. | 0.95 ✓ | 0.00 | 0.00 |
| **P2**: The Russian Civil War (November 1917 October 1922) was a multi-party war in the former Russian Empire immediately after the Russian Revolutions of 1917, as many factions vied to determine Russias political future. | 0.00 | 0.87 ✓ | 0.00 |
| **P3**: Alexander Fyodorovich Kerensky was a Russian lawyer and key political figure in the Russian Revolution of 1917. | 0.08 | 0.09 | 0.00 |

Table 14: Two examples from the HotpotQA distractor development set. Highlighted text shows the bridge entities for multi-hop reasoning, and also the words in red denote the predicted answer.

| | SQuAD Open | | Natural Questions Open | |
|---|---|---|---|---|
| | Retriever | Reader | Retriever | Reader |
| Avg. # of $L$ | 1.00 | 1.08 | 1.23 | 1.54 |
| 1 | 10,570 | 9,759 | 6,719 | 4,047 |
| 2 | 0 | 811 | 2,038 | 4,702 |
| 3 | 0 | 0 | 0 | 8 |

Table 15: **Statistics of the reasoning paths for SQuAD Open and Natural Questions Open:** the average length and the distribution of length of the reasoning paths selected by our retriever and reader for SQuAD Open and Natural Questions Open.

outputting [EOS]. In 588 examples (7.9%) of the entire distractor development dataset, the paragraph selection by our graph-based recurrent retriever differs from the top-2 strategy.

We present another example, where only the graph-based recurrent retrieval model succeeds in finding the correct paragraph pair, (**P1**, **P2**). The second question in Table 14 shows that at the first time step our retriever successfully selects **P1**, but does not pay attention to **P2** at all, as the retriever is not aware of the bridge entity, "the Russian Civil War". Again, once it is conditioned on **P1**, which includes the bridge entity, it can select **P2** at the second time step. Like this, we can see how our model successfully learns to model relationships between paragraphs for multi-hop reasoning.

# D    ADDITIONAL RESULTS ON SQUAD OPEN AND NATURAL QUESTIONS OPEN

Although the main focus of this work is on multi-hop open-domain QA, our framework shows competitive performance on the two open-domain QA datasets, SQuAD Open and Natural Questions Open. Both of the two dataets are originally created by assigning a single ground-truth paragraph for each question, and in that sense, our framework is not specific to multi-hop reasoning tasks. In this section, we further analyze our experimental results on the two datasets.

**SQuAD Open**    Table 15 shows statistics of the lengths of the selected reasoning paths on our SQuAD Open experiment. This table is analogous to Table 9 on our HotpotQA experiments. We can clearly see that our recurrent retriever always outputs a single paragraph for each question, if we only use the top-1 predictions. This is because our retriever model for this dataset is trained with the single-paragraph annotations. Our beam search can find longer reasoning paths, and as a result, the re-ranking process in our reader model somtimes selects the reasoning paths including two paragraphs. The trend is consistent with that in Table 9. However, the effects of selecting more than one paragraph do not have a big impact; we observed only 0.1% F1/EM improvement over our method with restricting the path length to one (based on the same experiment with $L = 1$ in Table 8). Considering that SQuAD is a single-hop QA dataset, the result matches our intuition.

**Natural Questions Open**    Table 15 also shows the results on Natural Questions Open, where we see the same trend again. Thanks to the ground-truth path augmentation technique, our recurrent retriever model prefers longer reasoning paths than those on SQuAD Open. We observed 1% EM improvement over the $L = 1$ baseline on Natural Questions Open, and next we show an example to discuss why our reasoning path approach can be effective on this dataset.

Table 16 shows one example where our model finds a multi-hop reasoning path effectively in Natural Questions Open (development set). The question "who sang the original version of killing me so" has relatively fewer lexical overlap with the originally annotated paragraph (**Killing Me Softly with His Song (V)** in Table 16). Moreover, there are several entities named as "killing me softly" in Wikipedia, because many artists cover the song. To answer this question correctly, our retriever first selects **Roberta Flack (I)**, and then hops to the originally annotated paragraph, **Killing Me Softly with His Song (V)**. Our reader further verifies this reasoning path and extracts the correct answer from **Killing Me Softly with His Song (V)**. This example shows that even without gold

---

**Q**: who sang the original version of killing me softly

---

**Roberta Flack (I)**: Roberta Cleopatra Flack (born February 10, 1937) is an American singer. She is known for her No. 1 singles "The First Time Ever I Saw Your Face", "Killing Me Softly with His Song"...

- - - - - - - - - - - - - - - - - - - - - - - - - - - - - - - - - - - - - - - - - - - - - - - - -

**Killing Me Softly with His Song (V)**, The song was written in collaboration with Lori Lieberman, who recorded the song in late 1971. In 1973 it became a number - one hit in the US and Canada for Roberta Flack, Many artists have covered the song....

- - - - - - - - - - - - - - - - - - - - - - - - - - - - - - - - - - - - - - - - - - - - - - - - -

Annotated reasoning path: **Killing Me Softly with His Song (V)**
Predicted reasoning Path: **Roberta Flack (I) → Killing Me Softly with His Song (V)**

---

Table 16: An example from Natural Questions Open. The bold text represents titles and paragraph indices (e.g., (I) denotes that the paragraph is an introductory paragraph). The highlighted phrase represents a bridge entity and the text in red represents an answer span.

reasoning paths annotations, our model trained on the augmented examples learns to retrieve multi-hop reasoning paths from the entire Wikipedia.

These detailed experimental results on the two other open-domain QA datasets demonstrate that our framework learns to retrieve reasoning paths flexibly with evidence sufficient to answer a given question, according to each dataset's nature.

