# OpenReview forum: "Learning to Retrieve Reasoning Paths over Wikipedia Graph for Question Answering"
_ICLR.cc/2020/Conference — Accept (Poster)_

### Official Review · AnonReviewer3 · 2019-10-22
**Official Blind Review #3**

**Rating:** 8

**Review:**

This paper proposes a method to find a sequence of reasoning paragraphs in Wikipedia to answer queries requiring multi-hop reasoning. They make the key observation that answering multi-hop queries might require retrieving evidence that have very less lexical overlap with the question. Given a query, the proposed method starts from a set of initial paragraphs retrieved by a tf-idf retriever and uses the outgoing Wikipedia anchor link to hop to the next evidence. They propose a simple recurrent neural network that takes in the current paragraph (and the hidden state) and decide which paragraph to hop to in the next step. Because of the available supervision for the paragraphs (in HotpotQA), they can train a supervised path selector. They also add a special EoE token that denotes the end of the reasoning path, thereby having the ability to produce reasoning paths of different lengths. After training the retriever a beam of reasoning paths is sent to the reader module. The reader module re-ranks the reasoning paths again and then use a standard BERTQA model and the top re-ranked chain of paragraphs to find the evidence.

Overall, the paper presents a well-designed system for handling multi-hop queries and the explicit recurrent state is a nice contribution and addition to the IR model proposed in Godbole et al., 2019. The paper is clearly written for the most part.

===Update (11/12/2019)===
The authors have addressed all my comments and have improved the results since. I am recommending acceptance. Nice work.

Strengths:
— The proposed method has demonstrated strong results on 2 datasets in challenging open-domain settings. The ablation results are helpful.
— The paper is clearly written and was straightforward to follow

Weaknesses:

1.  The paper mentions that it studies the interplay between the retriever and reader. It is unclear how it is doing so, since the retriever and the reader are not explicitly interacting with each other. Cant the retriever and the reader be trained separately?
2.  It is unclear / not motivated, why there is an extra step of re-ranking required in the reading stage? In other words, what kinds of extra inductive bias is this additional step of re-ranking providing since the same kind of supervision was used while training the retriever model. I do note that the ablation study is helpful and it is clear that it is effective, but it would be nice to see a discussion regarding why this second step of re-ranking helps.
3. Since the reader model (BERT reader) takes the top scoring chain of paragraphs concatenated together, that would imply that it is currently limited by the number of positional embeddings in the BERT model (512 tokens). I think this limitation should be explicitly mentioned and possible remedies discussed.
4. The current approach is heavily dependent on Wikipedia graph and will not work if the hyperlink graph is not provided. It would have been nice to have an entity linker component that could also create the graph structure. I believe concurrent work such as Godbole et al., 2019 has addressed this and the paper should mention this while contrasting with their work.
5. From figure 2, I got an impression that since the reader scored the span in "Top 2 reasoning path” higher, that was selected. But after section 3.2, I was left confused because it looks like the reader model consumes the top scoring chain after the second stage of re-ranking. This is not clear from the figure and should be fixed.
6. Discussion on scalability: Although the retriever is clearly very effective for such questions, the running time would be prohibitive (for open domain QA) as at test time, query dependent context representations is constructed for each of the paragraph in the reasoning chain. I would like to see a discussion / some running time comparison where query independent paragraph representations are constructed and the network just encodes the query independently at test time.

Minor: Typo liked -> linked (Sec 4.3, line 5)

**Experience Assessment:**

I have published one or two papers in this area.

**Review Assessment: Checking Correctness Of Derivations And Theory:**

N/A

**Review Assessment: Checking Correctness Of Experiments:**

I carefully checked the experiments.

**Review Assessment: Thoroughness In Paper Reading:**

I read the paper thoroughly.

---

> ### Author Response · Authors · 2019-11-12
> **Response to Official Blind Review #3 (part 1)**
>
> We really appreciate your supportive comments on our paper and your detailed feedback. Below, we address the weakness.
>
> # On the training of reader and retriever (re: Weaknesses 1)
> Our reader and retriever are separately trained, and this paper does not explore joint learning. We used the term “interplay” to represent our reasoning path re-ranking framework where our reader verifies the retrieved reasoning paths produced by the beam search, instead of finalizing the path selection only by the retriever. Our training strategy for our reader uses not only ground-truth paragraphs but also negative examples to simulate irrelevant paths produced by our retriever. Joint learning is interesting future work; nevertheless, such a two-stage training strategy is worth investigating, considering our strong empirical results. In practice, another advantage is that the framework is flexible; for example, when better reader models are made available, we can easily leverage the advance, without re-training the retriever model. For this revision, we re-train our reader models for SQuAD Open and HotpotQA and leveraging these new models further advances the state-of-the-art results on the two datasets.
>
>
> # On the inductive bias and the differences in supervision (re: Weaknesses 1, 2)
> There are practical differences in training our retriever and reader models.
>
> The first difference is in paragraph interactions. Our retriever learns to capture the paragraph interactions through the BERT’s [CLS] representations, after independently encoding the paragraphs along with the question; this makes our sequential retrieval scalable to the open-domain scenario. By contrast, our reader model fully leverages the self-attention mechanism across the concatenated paragraphs in the retrieved reasoning paths; this is especially crucial for multi-hop reasoning as discussed in recent work (Wang et al., 2019a).
>
> The second difference is in supervision signals. Our retriever is trained to predict plausibility of the reasoning paths, without learning to answer the question. Our reader model also learns to predict the plausibility with the stronger paragraph interactions, and jointly learns to answer the question.
>
> In summary, our retriever is scalable, but the top-1 prediction is not always enough to fully capture multi-hop reasoning to answer the question. Therefore, we use our reader model for the additional re-ranking process to mitigate the uncertainty and make our framework robust. Table 9 shows the statistics of the re-ranking results, and one interesting observation is that our reader model prefers longer paths. We added an example in Figure 4 (and also in Table 12 in Appendix) where the re-ranking finds more convincing reasoning paths.
>
>
> # The token length limitations by BERT (re: Weaknesses 3)
> We investigated the statistics of the token length of the concatenated paragraphs in the selected reasoning path for HotpotQA full wiki. In summary, only 0.2% of the examples exceed 512 tokens (based on the BERT tokenization), and thus we expect that the influence of BERT’s maximum length limitation is marginal. We believe this is another benefit of our framework. By selecting the reasoning path, our model can effectively avoid handling many paragraphs in the encoding steps.
>
> # Reliance on hyperlink information and experiments with off-the-shelf entity linking components (re: Weaknesses 4, Update 3)
> Our updated manuscript presents a comparison of our framework with and without the given hyperlinks. In place of the hyperlinks, we used an off-the-shelf entity linking system. Please refer to the details of the experiments in Section 4.4 “The performance with an off-the-shelf entity linking.” Table 7 shows marginal performance drop even without the hyperlinks, still achieving the state of the art on HotpotQA full wiki. We would also like to mention that the existence of hyperlinks is common, especially when using documents on the Web, and our results suggest that using hyperlinks as well as entity links is promising. The most recent work (Anonymous, 2019; Nie et al., 2019) also relies on the hyperlinks, while our results perform better.
>
> [...continued in next post]

---

> > ### Author Response · Authors · 2019-11-12
> > **Response to Official Blind Review #3 (part 2)**
> >
> > # Clarification on Figure 2 (re: Weakness 5, Update 6)
> > Sorry for the confusion caused by our current figures. We first would like to clarify that our reader reads all of the eight reasoning paths in parallel and jointly predicts P_i^{start}, P_i^{end} and P(E|q) for each of the reasoning path. To determine the final answer, the reader selects a span (i,j) from E, whose P(E|q) is the highest (See Equation 7). In our previous version, we call the reasoning path re-ranking (“second stage of re-ranking” in your words) as “answer re-ranking”, which might cause confusion. In our updated version, we re-name this module as “reasoning path reranking” for clarification and update all of the figures and relevant section titles.
> >
> >
> > # Performance and running-time comparison with query-independent encoding (re: Weaknesses 6)
> > Appendix A.2 discussed the motivation of using our query-dependent paragraph encodings. In our preliminary experiments, we started from a query-independent model with the RNN, but we found that the retrieval accuracy was very low even on the HotpotQA distractor setting. This motivated us to use the query-dependent representations, with the help of the initial TF-IDF retrieval. Lee et al. (2019) also showed that their query-independent retrieval model performs poorly on datasets requiring entity-centric retrieval such as SQuAD Open. The query-independent encodings save computational costs on the BERT encodings, while introducing other engineering efforts like how to store and retrieve pre-computed representations (Seo et al., 2019). By contrast, we primarily put more weight on improving the accuracy, keeping our model scalable by introducing our efficient inference strategies such as the beam search.
> > For an additional experiment about this, please also refer to another thread "Updates on empirical results on query independent encoding."
> >
> > # Typo liked -> linked (Sec 4.3, line 5)
> > Thank you for pointing it out. We have fixed the typo.

---

> ### Author Response · Authors · 2019-11-13
> **Updates on empirical results on query independent encoding**
>
> [For the response to individual weaknesses, please read our response to Official Blind Review #3 (part 1 and part 2).]
>
> To show the importance of the query-dependent encodings in terms of accuracy, we conducted an experiment again with a query-independent variant of our approach on the HotpotQA development set. More specifically, our retriever model encodes paragraphs independently from their corresponding queries and sequentially retrieves paragraphs in the same manner as our proposed approach. For a fair comparison, we use the same reader model (BERT wwm) with the path re-ranking. We train the alternative model without using the data augmentation technique for quick experiments. For evaluation, we provide results on both the distractor and full wiki settings.
>
> The results are summarized in the table below:
>
>
>   encoding method   | full wiki QA F1  | full wiki QA EM  | distractor QA F1  | distractor QA EM
> --------------------------   |----------------------|-----------------------|------------------------|-----------------------
>  query-dependent     |          64.1           |         52.6             |           81.2              |           68.0
>  query-independent |          47.3            |         37.8             |           80.0              |           66.4
>
> For our query-dependent approach, the full wiki results correspond to “retriever, no link-based negatives” in Table 6, and the distractor results correspond to “Ours (Reader: BERT wwm)” Table 1. As seen in this table, the QA F1 and EM performance significantly deteriorate on the full wiki setting, which demonstrates the importance of the query-dependent encoding for complex and entity-centric open-domain question answering. This is the reason why we employ our query-dependent approach, and we achieve competitive results on the three datasets.
>
> We also found that the performance drop on the distractor setting is much smaller than that on the full wiki setting. This is due to its closed nature. In the distractor setting, we are given ten paragraphs and the two gold paragraphs are always included, which makes the retrieval task much easier than that in the full wiki setting. We have only 10 paragraphs for each question, and thus the number of the possible reasoning paths is quite limited. Therefore, our recurrent retriever model is likely to discover the gold reasoning paths by the beam search, and our reader model can select the gold paths by the robust re-ranking approach. To verify this assumption, we checked the P EM score as a retrieval accuracy in the distractor setting. If we only consider the top-1 path from the beam search, the P EM score of the query-independent model is 12% lower than that of our query-dependent model. However, if we consider all the reasoning paths produced by the beam search, the coverage of the gold paths is almost the same. As a result, our reader model can perform similarly with both the query-dependent/independent approaches. This additionally shows the robustness of our re-ranking approach.
>
> We are planning to add these experimental results in our next revision.

---

### Official Review · AnonReviewer2 · 2019-10-24
**Official Blind Review #2**

**Rating:** 6

**Review:**

Summary
========
This paper introduces a graph-based recurrent retrieval model for retrieving evidence documents in a multi-hop reasoning question answering task. The main idea is that (1) the graph formed by Wikipedia links between passages can be used as constraint for constructing reasoning chains, and (2) the joint encoding of the question and current passage can be used to retrieve a subsequent passage in the reasoning chain. The paper describes a model for implementing the above retrieval system, and how they jointly train with a reading comprehension model. They demonstrate the effectiveness of the system on HotPotQA, showing improvements over previously published models, and SQuaD-Open, showing competitive results.

Overall Comments
===============
The paper is an interesting, but incremental, improvement to the area of question answering. Overall, there are two main concerns about this work. First, while the results are somewhat strong, the ideas presented are small variations on existing systems. For example, Godbole et al 2019 and Ding et al. 2019 both explore using graphical structural to constraint iterative, multi-hop, retrieval. Also, Feldman et al 2019, describe an encoder based approach to encode question and paragraph context for iterative retrieval. Asides from smaller modeling differences (choice of RNN, training regime, BERT reader, etc.) to account for the difference in results, the main difference seems to be the joint training of the retrieval system with the reader. Secondly, the paper lacks clarity on some formal definitions and definition of the graph, making it hard to understand the content precisely.

Detailed Comments
================
Below are some detailed comments about specific parts of the paper, in order of importance:

1. One important limitation of this technique is the reliance on a linked documents for constructing the retrieval system. It is not clear from the paper how much of the results are obtained from constraining the set of retrieved passages (after the initial retrieval) to Wikipedia links. And whether, for example, substituting Wikipedia links with links derived from an off-the-shelf entity linking system would suffice.

2. Given that the retrieval model is restricted to link structure in Wikipedia that induces the proposed retrieval graph, I assume that there are “reasoning paths” that do not exist in the graph, given Wikipedia’s policy of avoiding adding redundant links within a Wikipedia page. It would have been informative to conduct an “Oracle” experiment: that is, given the initial  set of retrieved nodes and the graph structure, are there *any* paths that provide the correct answer and reasoning chain? That is to say, what is the upper-bound performance on the proposed system given the currently induced Wikipedia graph?

3. In Section 3, and even later on in the paper, it was not clear what “E” denotes. It never seems to be defined, and is used interchangeably with “graph node”, “wikipedia page”, “wikipedia paragraph” and “reasoning path”. Are these the same thing? It would be much clearer to define what E means, and perhaps separate the different concepts (node, passage, reasoning path) properly.

4. In Section 3, it seems that ‘q’ is not defined. Is it the question?

5. In Section 3.1, it is not clear what the graph actually contains. Does it contain all the paragraphs from Wikipedia? Just the paragraphs with links? The first paragraph of every Wikipedia page? What granularity of the wikipedia page becomes an individual node in the graph?

6. In Section 3.1.1., the representation of the starting retrieval (i.e., time-step = 0), h_0, is not defined. Later in the section, the paper mentions the use of TF-IDF for the initial set of nodes, instead of the learned retrieval model. This seems a bit unusual design decision without further explanation. Particularly when taking the results in Table 4, showing TF-IDF based retrieval performs worse that the learned retrieval system from the proposed model.

7. In Section 3, C_{t} (the candidate set of paragraphs) is not defined. This is an important set to define. Is it the set of paragraphs derived from Wikipedia links, starting from the current node?

8. In Section 3.1.2, “Loss function”, the term g_{r} is not defined.

9. In Section 4.4 “Analysis on reasoning path length”, it would have been useful to see the performance of the model with different path lengths. This analysis is somewhat common on multi-hop reasoning tasks, and should be included.

10. Typo in Section 4.4: “..., and out model is likely too terminate …”  should be “ likely to terminate “


**Experience Assessment:**

I have read many papers in this area.

**Review Assessment: Checking Correctness Of Derivations And Theory:**

I assessed the sensibility of the derivations and theory.

**Review Assessment: Checking Correctness Of Experiments:**

I carefully checked the experiments.

**Review Assessment: Thoroughness In Paper Reading:**

I read the paper at least twice and used my best judgement in assessing the paper.

---

> ### Author Response · Authors · 2019-11-12
> **Response to Official Blind Review #2 (part 1)**
>
> We thank you for your helpful feedback. We have substantially updated our manuscript to address all the concerns you kindly raised as much as we can.
>
> First of all, we would like to address the two weaknesses you mention in your overall comments.
>
> Originality.
>
> # On the difference with other graph-based approaches
> Our work has several significant originalities in its system design, training and inference time strategies from Ding et al. (2019) and Godbole et al., (2019), which leads to more than 20 point improvements over these previous approaches on HotpotQA full wiki.
>
> 1)  System design: We formulate the retrieval as reasoning path search over the Wikipedia graph, instead of dynamically constructing an entity-graph for each question based on compiled document lists as in the previous work; the recurrent module dynamically updates and expands candidate paragraphs from the initial TF-IDF-based candidates at each time step. In addition, our work also studies the interplay between our retrieval model and the reader model (See the response of “Reasoning path retrieval and the interplay between our reader and retriever” below for details).
>
> 2) Training strategy: To train our recurrent module to learn to retrieve the path, leveraging the graph structure, we train our model with negative sampling and multiple reference paths (See Section 3.1.2 and the summary by review#1).
>
> 3) Inference strategy: We introduce beam-search based decoding to make the framework more scalable (See Section 3.1.1; also summarized by review#1 and #2), and the beam search with our reasoning path re-ranking is more effective than a greedy search. As in Table 6, replacing beam search with greedy search deteriorates F1 by 3.7. Also, our method does not need to encode all possible nodes like the previous studies, and instead each path only encodes its corresponding paragraphs.
>
> The HotpotQA dataset used in the previous work is based on introductory paragraphs only. By contrast, our method is applied not only to HotpotQA but also to the Natural Questions (See Update 1) and SQuAD Open datasets. These two datasets are not restricted to the introductory paragraphs, and our method achieves state-of-the-art results. This is made possible by our search-based decoding strategy. One interesting observation on our Natural Questions experiments is that our model learns to retrieve multi-hop reasoning paths with our training strategy, even without multi-hop gold path annotations as in HotpotQA (See Appendix C.5 and Table 13). This demonstrates the robustness and scalability of our approach.
>
>
> # On the difference with other multi-step retrieval approaches
> Previous multi-step approaches such as Das et al. (2019), Qi et al. (2019), Godbole et al. (2019) and Feldman and El-Yaniv (2019) do not accommodate arbitrary steps of reasoning. As review#1 and review#3 summarize, our RNN approach uses the EOE symbol to produce reasoning paths with different lengths. This allows our model to be easily applicable to both multi-hop and single-hop questions without specifically changing the model architecture. Table 8 demonstrates the effectiveness of this adaptive retrieval process. In practice, it is not obvious if a question requires single-hop or multi-hop retrieval (e.g., some of the Natural Questions Open are clearly answerable based on single paragraph, while in some questions, multi-hop reasoning helps.), and thus this flexibility is another significant advantage.
>
>
> # Reasoning path retrieval and the interplay between our reader and retriever
> Our framework benefits from the interplay between our retriever and reader. Our retriever encodes the candidate paragraphs independently for scalability, and iteratively selects a paragraph at each time step conditioned by the prediction history. Each of the resulting K reasoning paths (K=beam size) includes one or more paragraphs. Our reader encodes the paragraphs in the paths jointly and predicts probabilities of each reasoning path E containing an answer span. By encoding the paragraphs jointly, our reader model fully leverages the self-attention mechanism across the concatenated paragraphs in the retrieved reasoning paths; this is especially crucial for multi-hop reasoning as discussed in recent work (Wang et al., 2019a). The additional reasoning path re-ranking makes our overall framework robust, leading to large performance improvement (See Section 4.4 and Table 8,9). This reasoning path re-ranking is one of the novel points in our work.
>
> These significant differences together demonstrate state-of-the-art performance on the four experimental settings in the three datasets, HotpotQA (full wiki, distractor), SQuAD Open and Natural Questions Open (Update 1). Notably, our method outperforms all the previous graph-based or multi-step retrieval methods by more than 20 points on HotpotQA full wiki and 15 points on SQuAD Open (See Table 1,2,3).
>
> We added discussions in Section 2 (Related Work) to clarify these points.

---

> > ### Author Response · Authors · 2019-11-12
> > **Response to Official Blind Review #2 (part 2)**
> >
> > Clarity.
> >
> > Thank you for pointing out several unclear descriptions in our method, and in this revision, we did our utmost best to clarify some details and also add additional experimental results to make our results more convincing (See the details from the response on “The clarity on the definition” below).
> >
> > ================
> >
> > We list our response to your Detailed Comments below.
> >
> > # On the reliance on hyperlinks and performance evaluation with an off-the-shelf entity linking system (re: Detailed Comments 1)
> > Our updated manuscript presents a comparison of our framework with and without the given hyperlinks. In place of the hyperlinks, we used an off-the-shelf entity linking system. Please refer to the details of the experiments in Section 4.4 “The performance with an off-the-shelf entity linking.” Table 7 shows marginal performance drop even without the hyperlinks, still achieving the state of the art on HotpotQA full wiki. We would also like to mention that the existence of hyperlinks is common, especially when using documents on the Web, and our results suggest that using hyperlinks as well as entity links is promising. The most recent work (Anonymous, 2019; Nie et al., 2019) also relies on the hyperlinks, while our results perform better.
> >
> >
> > # The clarity on the definition of the Wikipedia graph, reasoning paths, paragraph candidates C_{t}, and search spaces (re: Detailed Comments 2, 3, 5, 7)
> > We have updated Section 3 (Overview) and Section 3.1 and 3.2 to clarify the definitions.
> >
> > - The Wikipedia Graph \mathcal{G}: each node of \mathcal{G} is “a paragraph”. A paragraph is represented as p_i in our paper. Thus, an edge connects two paragraphs, and it can be either a hyperlink or a with-in article link (See Section 3.1). By default we consider all of the paragraphs in English Wikipedia, but for HotpotQA, we only consider introductory paragraphs, following all the previous work using the dataset. This is described in Section 4.1 “Evidence Corpus and the Wikipedia graph”. The whole graph is constructed in advance, and we reuse the same graph for training and inference, instead of dynamically building entity graph everytime from the TF-IDF retrieval results as in Ding et al. (2019) or Godbole et al. (2019).
> >
> > - A reasoning path E = [p_i, \ldots p_k]: a reasoning path contains one or more paragraphs that are together used by out reader model to answer a given question. Our framework learns to retrieve a reasoning path for a given question from the entire Wikipedia.
> >
> > - Top B reasoning paths \mathbf{E}=\{E_1, \ldots, E_B\}: our retriever employs a beam search with the beam size of B for decoding, and thus, the top B distinct reasoning paths \{E_1, \ldots, E_B\} are returned. Our reader further re-ranks these reasoning paths to determine an answer. Our reader jointly encodes all of the paragraphs in each reasoning path and then re-ranks the retrieved reasoning paths, fully capturing the paragraph interactions in E.
> >
> > - The candidate paragraph set C_{t}: we construct C_{t} as a set of the paragraphs to be considered for retrieval at each time step t. At t=1, the candidates are initialized with the F paragraphs with the highest TF-IDF scores with respect to the question. Then, C_{t+1} includes (i) paragraphs linked from the previously selected paragraph p_{t} or (ii) a few of top ranked paragraphs from the previous step. C_{t} is not limited to the initially retrieved F paragraphs, and thus, our retriever dynamically expands the paragraph candidates over the graph.
> >
> >
> > # On the upper-bound performance of the proposed system (re: Detailed Comments 2)
> > We expect that your suggestion about “upper-bound performance” is to calculate how many of the ground-truth reasoning paths can be found in our initial TF-IDF retrieval and the graph. As shown in our experimental results on HotpotQA, a retrieved reasoning path consists of up to three paragraphs. Thus, we consider if a ground-truth reasoning path can be found in the subgraph within three steps from the initial TF-IDF retrieval.  In particular, we calculate the upper-bound paragraph EM as follows:
> > (the number of questions whose ground-truth reasoning paths can be found in the sub-graph) / (the total number of questions).
> >
> > We have checked the upper bound in our preliminary experiments on the development set in HotpotQA. The coverage of the gold reasoning paths is 75.4% when the initial TF-IDF retrieval size is 20. The coverage is considered as an upper bound of the P EM score of our method in Table 5. For reference, the coverage is 84.1% and 89.2% with the initial TF-IDF size of 100 and 500, respectively. This analysis is now reported in Appendix C.1. [...continued in next post]

---

> > > ### Author Response · Authors · 2019-11-12
> > > **Response to Official Blind Review #2 (part 3)**
> > >
> > > # On the effectiveness of TF-IDF-based initial paragraph candidates (re: Detailed Comments 6)
> > > We initialize the candidate paragraph set C_1 with TF-IDF-based top F paragraphs, and start using our RNN retriever from the candidate set. There are several reasons for this strategy.
> > >
> > > First, processing millions of paragraphs with neural networks is computationally infeasible, especially for non-industry scale computational resources, as discussed in previous work (Lee et al., 2019; Seo et al., 2019).
> > >
> > > Second, the previous work shows that fully trainable retrieval without using any term-based features performs poorly for entity-centric questions (e.g., SQuAD), as compressing specific entity information into a fixed dimensional vector is challenging (Seo et al., 2019, Lee et al., 2019). In particular, a recently proposed end-to-end retriever, ORQA (Lee et al., 2019) shows significantly lower performance than a TF-IDF retriever (DrQA proposed by Chen et al., 2017) on SQuAD Open (See Related Work and Table 3).
> > >
> > > For the reasons we listed above, we bootstrap the retrieval with the TF-IDF retriever. We clarified these points in Section 3.1.1 in our updated manuscripts.
> > >
> > >
> > > # The definition of q (re: Detailed Comments 4)
> > > We have updated the manuscript to clear define q as a question.
> > >
> > >
> > > # The initialization of h_0 (re: Detailed Comments 6)
> > > Our RNN’s initial state is h_1, which is used to predict the first paragraph in each reasoning path. h_1 is based on an independent parameterized vector, and please also refer to Appendix A.1 for more details.
> > >
> > >
> > > # The definition of “Loss function”, the term g_{r}  (re: Detailed Comments 8)
> > > We have found that ``_{r}’’ was missing from the definition of ``g_{r}’’ in the sentence: ``In particular, we add a new training path g = [pr, p1, . . . , p|g| ] by …’’ in Section 3.1.2. We have revised this part to precisely define the term.
> > >
> > >
> > > # The performance with different path lengths (re: Detailed Comments 9)
> > > We have added the performance comparison with the settings where we use the same model but set the length of the reasoning paths to a fixed number (i.e., 1, 2, 3, 4). We can see that our adaptive approach performs the best, even though the HotpotQA’s gold reasoning path length is always 2 (See Table 8).
> > >
> > > We further present the QA performance of the model with different lengths (averaged QA F1 and EM scores on the questions whose retrieved reasoning path length is {1,2,3}) on HotpotQA (See Table 9).
> > >
> > >
> > > # A typo in Section 4.4 (re: Detailed Comments 10)
> > > Thank you for pointing it out. We fixed this typo in our updated submission.

---

### Official Review · AnonReviewer1 · 2019-10-24
**Official Blind Review #1**

**Rating:** 6

**Review:**

The paper is proposing a multi-hop machine reading method tested on hotpotqa in the Full Wikipedia setting and squad-open datasets.
For hotpotqa, It could also have been interesting to evaluate the method of the distractor ones.
First, the proposed method constructs a graph over the Wikipedia pages represented by their respective summary paragraphs.
In this representation, the hyperlinks among pages represent the edges.
Then, the authors trained a normalized RNN model to retrieve the candidate reasoning paths from the question.
The model is bootstrap using TF-IDF page retrieval techniques.
Then, a Beam-search decoding strategy is used to retrieve "reasoning path" which is then pass through a BertQA model using a simple question-reasoning-path concatenation technique.
One originality of the method is the negative sampling strategy that includes negative TF-IDF retrieval as starting points to robustify the sequential extraction process.
The detailed experiments and ablation tests give to illustrate the experimental relevance of the proposed method.

**Experience Assessment:**

I have published in this field for several years.

**Review Assessment: Checking Correctness Of Derivations And Theory:**

I carefully checked the derivations and theory.

**Review Assessment: Checking Correctness Of Experiments:**

I carefully checked the experiments.

**Review Assessment: Thoroughness In Paper Reading:**

I read the paper thoroughly.

---

> ### Author Response · Authors · 2019-11-12
> **Response to Official Blind Review#1**
>
> Thank you for reading our paper thoroughly and providing encouraging feedback.
>
> # The results on HotpotQA distractor setting
> Regarding the Hotpot distractor setting, we have evaluated our method on the settings, and the scores on the development set are reported in the first version of our manuscript (See Table 1, columns 6-9); due to the time constraints, we did not submit our model to the leaderboard. The results show that our method achieves state-of-the-art scores on the distractor setting, outperforming the previous best-published model by more than 10 points. Our work is the first to demonstrate the state-of-the-art performance on both the distractor and full wiki settings of HotpotQA. We revised our manuscript to make the distractor evaluation clear (See Update 7 and Section 4.1 “HotpotQA” and  Section 4.2 in our updated manuscript). We have a qualitative example in Appendix C.6 and Table 14, which shows how our sequential reasoning path process also helps in distractor setting.
>
> We added new experimental results on Natural Questions Open (Table 4) and additional analysis on selected reasoning paths (Section 4.4) in our updated version. We hope it will be helpful in evaluating the effectiveness of our method.

---

### Public Comment · ~Siqi_Sun2 · 2019-10-05
**question about table 4 (Retrieval evaluation)**

First of all, thanks for posting your amazing work here! The paper's novelty, presentation and results are all excellent to me!

However we do have one question about your results in table 4, may I ask how do you compute the AR and PR (especially PR) in table 4 (some rows are copied below) ? Note that we absolutely trust your results because they are evaluated on a hidden test set, we just couldn't reproduce the numbers in this specific table and hope you could help if possible :)

Models                        AR       PR      PEM       EM
--------------------------------------------------------------------
TF-IDF                         39.7    66.9    10.0       18.2
Cognitive Graph       76.0    87.6     57.8      37.6

For us we just use PR = number of retrieved gold paragraph /  total number of gold paragraph, then average PR across different questions. The numbers in table confuse me because
(1) The PR for TF-IDF on dev set is way too high. The hits@10 for TF-IDF retriever in HotpotQA paper is 56.06 on dev, while your TF-IDF achieves 66.9 recall on top 2...  We also implemented our own TF-IDF retriever and achieved 55.71 recall if we retrieved 10 paragraphs per question, which is similar to original HotpotQA paper (of course, way below your number, especially considering your model only retrieves top 2)

(2) We ran the CogQA retriever and achieved 69.98 PR (in table 4 the number is 87.6). Since the retrieved documents should be almost the same for different runs, can you help us find out what might be the reason that we failed to replicate the results?

Thanks again for your time.

---

> ### Author Response · Authors · 2019-10-07
> **Re: question about table 4 (Retrieval evaluation)**
>
> Hi Siqi,
>
> Thank you for your interest in our work and such an encouraging comment. We agree that our current description of paragraph recall (PR) might be not clear enough, and we will update our manuscript once the discussion phase begins and edits are allowed. In summary, our PR evaluates the percentage of questions for which at least one gold paragraph is retrieved. We additionally contrast our PR metric with PEM, which evaluates the percentage of questions for which both gold paragraphs are retrieved.
>
> > For us we just use PR = number of retrieved gold paragraph /  total number of gold paragraph, then average PR across different questions.
> >  (1) The PR for TF-IDF on dev set is way too high. The hits@10 for TF-IDF retriever in HotpotQA paper is 56.06 on dev, while your TF-IDF achieves 66.9 recall on top 2...  We also implemented our own TF-IDF retriever and achieved 55.71 recall if we retrieved 10 paragraphs per question, which is similar to original HotpotQA paper
>
> By PR, we evaluate if at least one of the ground-truth paragraphs for each question is included among the retrieved paragraphs. In particular, the score is calculated as follows:
>
> PR = (the number of questions where a retriever finds at least one of the gold paragraphs) / (the total number of questions in the development set).
>
> We assume that the PR would be estimated higher than the recall score calculated by you or the hits@10 calculated by the HotpotQA authors.
>
> > (2) We ran the CogQA retriever and achieved 69.98 PR (in table 4 the number is 87.6). Since the retrieved documents should be almost the same for different runs, can you help us find out what might be the reason that we failed to replicate the results?
>
> We expect that this happens due to the same reason we described above.
>
> In addition, we would like to mention the motivations behind the metric design. Here, we aim at evaluating (1) if a retriever can find at least one paragraph of the gold paragraphs (corresponding to PR), and (2) if a retriever can find all of the gold paragraphs  (corresponding to P EM). We expect that even a non-parameterized retriever (e.g., TF-IDF retriever) can find one of the gold paragraphs based on lexical matching, but it is likely to fail to find one or more of the gold paragraphs consisting of little lexical overlap or semantic relationship to the original questions.
>
> As in Table 4, the PR is relatively high across several retrievers, but the TF-IDF retriever or the Re-rank show low P EM, as it cannot access the paragraphs that are ranked lower by TF-IDF but entailed or the relationships between paragraphs.
>
> We will add more detailed explanation about how we calculate the scores and why we design the metrics in the way in our next version. We also consider changing the names (PR and P EM).
>
> Again, thank you so much for your interest and insightful comments.

---

> > ### Public Comment · ~Siqi_Sun2 · 2019-10-07
> > **thanks for your clarification**
> >
> > Huge thanks for your detailed and valuable information, they are extremely helpful, much appreciated!
> >
> > With your explanation, all numbers make sense to me now, thanks a lot for your time!

---

### Author Response · Authors · 2019-11-12
**Summary of general updates**

We thank all of the reviewers for providing such insightful and valuable feedback. Based on the feedback, we made substantial updates on our paper.

Our updates are summarized below:

[Update 1] New experimental results on Natural Questions Open:
We added new experimental results on Natural Questions Open (Lee et al., 2019) to show our method’s robustness and scalability. Natural Questions Open has three unique features as an open-domain QA benchmark. The questions are written by actual users independently from existing corpora. Some questions in this dataset require multi-hop reasoning (e.g., how tall is the actor who plays hagrid in harry potter), but the multi-hop reasoning annotations are not provided. This dataset requires a system to search *all* paragraphs in Wikipedia articles, and thus a system needs to be truly scalable. Our results are competitive with the state of the art (See Section 4.2 and Table 4), which demonstrates the scalability and robustness of our framework. We also provide a qualitative example which shows that our system learns to conduct multi-hop retrieval on Natural Questions Open without original multi-hop reasoning path annotations (See Appendix C.5 and Table 13). We do not add any architectural design changes for this experiment.

[Update 2] Updated results on HotpotQA (distractor, full wiki) and SQuAD Open:
We found that our reader model was under-tuned on HotpotQA and SQuAD Open during our experiments on Natural Questions Open. In particular, it is effective to use larger mini batches for training our reader model with BERT and distant supervised examples extracted from our training data for the retriever model (See Section 3.2 and Appendix B.3). Consequently, we advance our state-of-the-art results from our initial submission on HotpotQA (both fullwiki and distractor) by around 4 points and also outperforms the state of the art (multi-passage BERT) on SQuAD Open by 3 points (See Section 4.2 and Table 1,3). We re-submitted to the HotpotQA full wiki leaderboard on November 6th ( https://hotpotqa.github.io/ ), and our model ranks first, outperforming all of the published and up-to-date unpublished work (See Table 2).

[Update 3] Performance using an off-the-shelf entity linking system vs. hyperlinks (review#2, review#3):
We added an experiment by replacing the Wikipedia hyperlinks with entity links given by an off-the-shelf system, and observed a slight performance drop (2.3 F1 and 2.2 EM), still achieving the state of the art (See Section 4.4 "The performance with an off-the-shelf entity linking," Table 7, and Appendix B.6 for details). This also suggests that using hyperlinks is promising, considering that hyperlinks are commonly used on the Web.

[Update 4] Clarification of the method (review#2):
We added clarification about the definition of the Wikipedia graph, reasoning path and search space by our retriever in Section 3 Overview and Section 3.1.1.

[Update 5] Additional analysis of experimental results (review#2, #3):
We added analysis on (1) the performance comparison with different reasoning path length (See Section 4.4 “The performance of different path length” and Table 8) and (2) qualitative and quantitative analysis to understand the importance of the reader-side reasoning path re-ranking (See Section 4.4 “The effectiveness of the interplay between retriever and reader”, Table 9, Figure 4).

[Update 6] Rename “answer re-ranking” to “reasoning path re-ranking” (review#3):
We rename “answer re-ranking” to “reasoning path re-ranking” to reflect our framework’s actual behavior and to avoid confusion.

[Update 7] Clarify the experimental settings and results on HotpotQA distractor (review#1):
In our first version, we briefly described the experiments on HotpotQA distractor. We added descriptions of the experimental settings and results for this setting in Section 4.1 and 4.2.

It should also be noted that Godbole et al. (2019) presented their work in Machine Reading for Question Answering workshop in EMNLP 2019 on November 4th, and the original manuscript was submitted to arXiv on September 17th (one week before ICLR 2020 deadline). Nevertheless, we did our utmost efforts to provide a careful comparison, and empirically our approach yields more than 20 F1 and 30 P EM improvements over this work. In this revision, we also added comparison with the most up-to-date work on HotpotQA posted after the ICLR submission deadline (Qi et al., 2019; Anonymous, 2020; Nie et al., 2019).

---

### Author Response · Authors · 2019-12-21
**Summary of updates (Revised on December 20th, 2019)**

We thank the reviewers and the PCs for their insightful and helpful feedback. We have incorporated some experimental results and analyses that we show in the response to the reviewers as well as updated the figures presented in our paper.

[Update 1] Add performance comparison of query-dependent and query-Independent encoding
We add the performance comparison of query-dependent and query-independent encoding mentioned in our response to review#3 (https://openreview.net/forum?id=SJgVHkrYDH&noteId=BJegSqBYsH) to Appendix C.4 and also add the detailed modeling design of query-independent encoding in Appendix A.2.

[Update 2] Add detailed results on SQuAD Open and Natural Questions Open
We further analyze our experimental results on SQuAD Open and Natural Questions Open (Appendix D).

[Update 3] Update figures
We have updated Figures 1, 2, 3, and 4.

---

### Author Response · Authors · 2020-02-14
**Summary of updates (Revised on February 13th, 2020)**

We add a few updates for our camera-ready version.

[Update 1] Add a link to our official implementation.
We open-source our PyTorch code with all of the train datasets and processed Wikipedia databases (https://github.com/AkariAsai/learning_to_retrieve_reasoning_paths ). We add the link to this Github repository on the first page.

[Update 2] Add discussions on some recent related work
We incorporate some additional discussions on some recent work (e.g., PullNet by Sun et al., 2019).

---

### Author Response · Authors · 2020-05-19
**update (Revised on May 19th, 2020)**

We fixed a typo in Figure 2.

---

### Decision · Program_Chairs · 2019-12-19

**Decision:**

Accept (Poster)

**Comment:**

The paper proposed a multi-hop machine reading method for hotpotqa and squad-open datasets. The reviewers agreed that it is very interesting to learn to retrieve, and the paper presents an interesting solution. Some additional experiments as suggested by the reviewers will help improve the paper further.